# Age is an intrinsic driver of inflammatory responses to malaria

Jessica R. Loughland[1,9], Nicholas L. Dooley [1,9], Zuleima Pava[1], Arya SheelaNair[2], Dean W. Andrew[2], Peta Tipping[3], Peter Bourke[4], Christian R. Engwerda[2,5], J. Alejandro Lopez [2,5], Kim A. Piera [3], Timothy William[3,6,7], Bridget E. Barber [2,3,6], Matthew J. Grigg [3,6], Nicholas M. Anstey[3,6], Gabriela Minigo [3,8] & Michelle J. Boyle [1,3,5] ✉

Age is a critical factor in immune responses to infection. In malaria, severe disease risk increases with age in non-immune individuals. Malaria severity is in part driven by inflammation, but mechanisms contributing to age-dependent disease risk are incompletely understood. We assessed inflammatory cytokines during malaria in non-immune children and adults, and innate cell responses in vitro to malaria parasites in naive children and adults. We show during malaria age is associated with increased inflammatory chemokines CCL2, CCL3, CXCL8, CXCL9, along with CRP, and IDO, which associate with symptoms. In naive individuals, classical monocyte and Vδ2⁺ γδ T cells from adults have higher inflammatory cytokine production, and transcriptional activation following stimulation with parasites. Classical monocyte responses in adults are dominated by CCL2, while in children increased IL10 and enrichment of IL10 signaling pathways is detected. Findings identify age-dependent cellular mechanisms that play crucial roles in driving inflammatory responses in malaria.

Age is an intrinsic factor in the host response to infection, with implications for disease pathogenesis and severity[1,2]. A recent systematic analysis revealed that for many infections, disease severity follows a 'J' or 'U' curve with age, with a relatively high risk in infants followed by the lowest risk in older children, before severity risk increases again in adults and the elderly[3]. The immune mechanisms underpinning these age-dependent changes are largely unknown. For malaria, caused by *Plasmodium falciparum* parasite infection, the global clinical burden predominantly occurs in children, who experience the highest risk of severe disease[3]. In high-transmission endemic areas, children are repeatedly infected before developing immunity which protects into adulthood, with disease severity following an 'L' distribution[3–5]. However, in areas where malaria control has reduced

transmission intensity, the burden of severe disease shifts from infants to older children[6,7]. Further, in low-transmission regions with unstable malaria, exposure and clinical immunity in childhood is much less common, and clinical disease, severe malaria and fatal *falciparum* malaria is seen in both children and adults[8]. As seen for other infections, host age in these non-immune populations is an intrinsic factor in malaria pathogenesis and disease severity[9]. In malaria-naive migrant populations moving from malaria-free to endemic areas, risk of severe disease is higher in adults compared to children during initial infection[10,11]. Consistent with this, amongst patients with severe disease, risk of death is higher in adults compared to children[12]. Age-dependent mechanisms contributing to the risk of severe malaria are incompletely understood.

[1]Burnet Institute, Melbourne, VIC, Australia. [2]QIMR Berghofer Medical Research Institute, Brisbane, QLD, Australia. [3]Menzies School of Health Research, Charles Darwin University, Darwin, NT, Australia. [4]Division of Medicine, Cairns Hospital, Manuda, QLD, Australia. [5]School of Environment and Sciences, Griffith University, Griffith, QLD, Australia. [6]Infectious Diseases Society Kota Kinabalu Sabah-Menzies School of Health Research Program, Kota Kinabalu, Sabah, Malaysia. [7]Subang Jaya Medical Centre, Selangor, Malaysia. [8]Faculty of Health, Charles Darwin University, Darwin, NT, Australia. [9]These authors contributed equally: Jessica R. Loughland, Nicholas L. Dooley. ✉e-mail: michelle.boyle@burnet.edu.au

Acquisition of adaptive immunity increases with exposure, concurrently with age, complicating our ability to identify age-intrinsic mechanisms of disease risk. Indeed, few studies have compared innate immune inflammatory responses in children and adults with naturally acquired *falciparum* malaria, which is only possible in low transmission areas where immunity is also low, and all ages are susceptible to malaria. Clinical symptoms of malaria occur during the blood stage of parasite infection, where activation of immune cells and production of inflammatory mediators contribute to disease[13]. Important innate immune cell responders include classical monocytes[14], the innate lymphocyte γδ T cell subset expressing Vγ9Vδ2 TCR (Vδ2 T cells)[15], and natural killer (NK) cells[16]. These cells, particularly monocytes and Vδ2 T cells, produce key inflammatory cytokines which are associated with severe disease[17], and control of these inflammatory responses is associated with tolerogenic immunity[18]. For example, in immune Malian adults there is an expansion of regulatory monocytes which produce IL10 in response to malaria[19]. This expansion is malaria-driven, with Malian children having monocytes phenotypically and functionally similar to malaria-naive adults[19]. However, no malaria-naive children were included in this study, so the influence of age on the monocyte response to malaria parasites is unknown. Similarly, reduced risk of malaria disease following repeat *falciparum* infection is associated with a reduction in the inflammatory cytokine response by γδ T cells[20,21]. However, age-associated changes to Vδ2+ γδ T cells have also been reported, with increased inflammatory and cytotoxic potential expanding markedly after birth[22–24]. The impact of these age-driven changes on the ability of malaria-naive individuals to respond to *Plasmodium* parasites is unknown. For NK cells, exposure to malaria is also linked to the expansion of adaptive rather that innate likes subsets[25], but whether age also impacts these changes or responses to parasites is unknown.

Here, leveraging a unique cohort of children and adults with *falciparum* malaria from a pre-elimination low transmission area[26], we identify inflammatory cytokines which are associated with age and clinical symptoms. To investigate cell mechanisms contributing to age-dependent inflammation during malaria, we focus on the major innate sources of inflammatory cytokines[17] and examine monocyte and Vδ2+ γδ T cell responses to parasite stimulation in vitro using cells from malaria-naive children and adults. Together, these data demonstrate the impact of host age on the innate immune cell responses to *P. falciparum* parasites.

## Results

### Inflammatory cytokine levels correlate with age in naturally acquired malaria

Malaria disease severity, partly driven by inflammation, is associated with age in non-immune populations[10–12]. In malaria endemic areas with high transmission, immunity develops with age, and adults rarely experience symptomatic disease[4]. To examine the effect of age on malaria-induced inflammation during natural infection, we analyzed inflammatory markers in plasma samples collected from children and adults who were enrolled in previously completed studies in a low malaria transmission area in Malaysia, where all ages remain susceptible to disease due to the limited prior exposure[26] (total *n* = 97, 78 male (80%), age 21 (median) [IQR 15–45 range 2–72] years, *n* = 79 from patients presenting with malaria at Kudat Division district hospital[26], and *n* = 18 enrolled at a tertiary referral hospital for the West Coast and Kudat division[27], Supplementary Table S1). Amongst malaria patients, 73 (75.3%) had uncomplicated malaria, with the remainder having severe disease (defined using World Health Organization 2014 research criteria). Amongst malaria patients in this sample set, parasitemia was not associated with age (Fig. 1a, *rho* = 0.055, *p* = 0.59). Plasma concentrations of 18 analytes related to immune inflammation and activation were analyzed, revealing a positive correlation among inflammatory markers CCL2, CCL3, CCL4, CXCL8, CXCL9, CXCL10,

CXCL11, as well as IDO, TRAIL, OPG and IL10 (Fig. 1b). Several of these inflammatory cytokines also correlated with parasitemia, consistent with a role of parasite burden in inflammation and disease (Fig. 1b). Age was significantly correlated with CRP, CCL2 (MCP-1), CCL3, CXCL8, CXLC9, and IDO (Fig. 1b/c). Age associations were not driven by severe disease, with all correlations remaining significant when considering only patients with uncomplicated malaria (Supplementary Fig. 1a). Further, age was not associated with fever duration prior to presentation (rho 0.015, *p* = 0.89), suggesting that associations were not confounded by prior duration of disease which may differ between children and adults. To assess if age-dependent inflammatory cytokines may contribute to disease, associations between disease symptoms and analytes were explored. Inflammatory analytes that were significantly associated with age, were also higher in patients with symptoms of rigors (Fig. 1d), myalgia (muscle pain) (Fig. 1e), headache (Supplementary Fig. 1b) and arthralgia (joint pain) (Supplementary Fig. 1c). CRP was significantly higher in patients with these symptoms, while chemokines CCL2 (MCP-1), CCL3 (MIP-1α) and CXCL8 (IL-8) were higher in patients with rigors and myalgia (Supplementary Fig. 1b–d). These relationships with symptoms were maintained when only analyzing patients >12 years of age, to account for potential reduced self-reporting of symptoms in children (Supplementary Fig. 1d). We also examined age relationships with objective clinical findings. Serum alanine aminotransferase (ALT) data was available for 62 individuals (64%) and was weakly associated with increasing age (*rho* = 0.24, *p* = 0.06) and CCL2 (*rho* = 0.22, *p* = 0.081) (Supplementary Fig. 2). Elevated ALT, a clinical measure of liver function, has been linked to parasite load and inflammatory responses during uncomplicated malaria caused by *P. falciparum*[28] and *Plasmodium vivax*[29] in populations with little pre-existing immunity. However, these findings suggest that patient age may also play a significant role in ALT elevations in non-immune individuals. Finally, we re-analyzed all clinical data from the district hospital cohort, previously compared to other malaria species infections[26], for differences in clinical presentation based on age (Supplementary Table S2). Among symptoms at enrollment, rigors, headache, myalgia and arthralgia were more frequent in adults consistent with increased inflammation and associations between inflammatory markers and symptoms. However, as mentioned this may be confounded by higher reported symptoms in adults. The objective clinical measure of oxygen saturation was also lower in adults, consistent with increased clinical disease with age. Duration of fever prior to presentation did not differ between children and adults (Supplementary Table S2). Together data highlight the relationship between age, inflammation and disease in malaria.

### Inflammatory cytokine production in monocytes following malaria stimulation is higher in naive adults

To understand cellular responses that may mediate increased inflammation during malaria, we analyzed phenotypes, functions and transcriptional activation profiles of immune cells relevant to malaria responsiveness in a cohort of malaria-naive individuals (children, <12 years, *n* = 13, age 8 [3–12] years (median[IQR]), 38% female; adults *n* = 13, age 42 [29–46] years (median[IQR]), 54% female, Supplementary Table S3). Analysis of ex vivo cell frequencies and phenotypes identified several differences, including higher proportions of CD14+ (classical) monocytes, CD16+ (non-classical) monocytes, classical dendritic cells (cDCs) and natural killer cells (NKs) in adults and a higher proportion of Vδ2+ γδ T cells in children (Supplementary Fig. 3a–e). Among CD4+ T cells, adults had a higher proportion of T-follicular helper (Tfh) cells, while other CD4+ effector T cells and FoxP3+ regulatory T cell (Treg) proportions were similar (Supplementary Fig. 3d, e). Age dependent differences in expression of markers associated with activation and function (HLA-DR, CD86, CD16 and ICOS) were also detected (Supplementary Fig. 3f–h). There was increased MHC-class II (HLA-DR) and Fc receptor (CD16) expression by classical DCs in adults

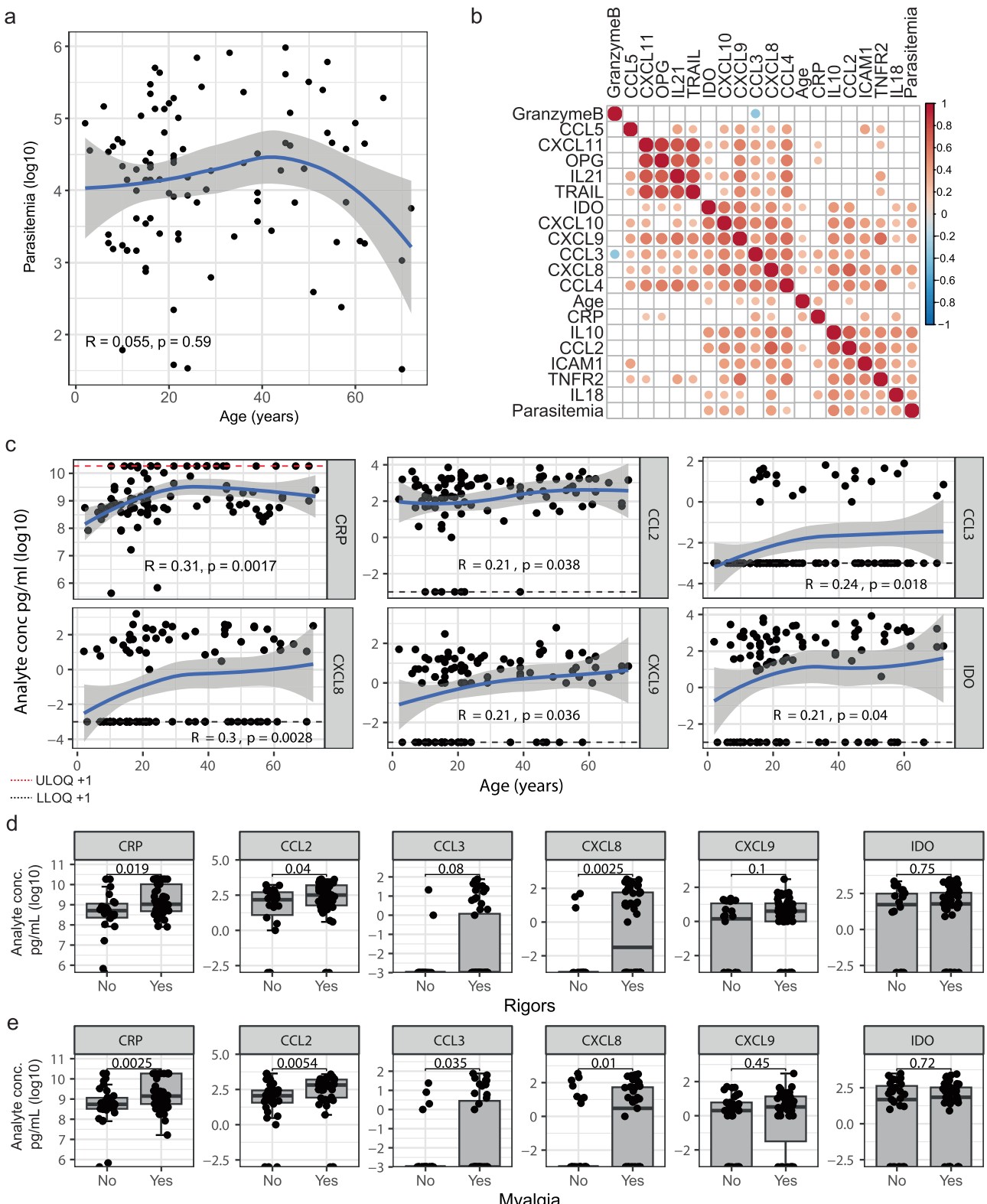

**Fig. 1 | Inflammatory cytokines increase with age in patients with malaria.**
Thirteen analytes were measured in plasma from 97 individuals with acute malaria (78 male (80%), age 21[15–45] years (median [IQR])). **a** Scatter plot of parasitemia (log10 parasites/μL) and age (years). Solid lines are LOESS fit curves with error bands of 95% confidence interval. Spearman's *Rho* and p are indicated. **b** Correlation matrix of inflammatory cytokines (pg/mL(log10)), age (years) and parasitemia (log10 parasites/μL). Correlations with *p* > 0.05 are blank and point size scales with Spearman's Rho. **c** Individual correlation plots of analytes with significant associations with age. Dashed Horizontal lines represent upper limit of

quantification plus one (red: ULOQ +1) and lower limit of quantification minus one (black: LLOQ −1). Solid lines are LOESS fit curves with error bands of 95% confidence interval. Analyte concentration for patients with and without **d** rigors or **e** myalgia (muscle pain). Dots represent individuals. Spearman's *Rho* and *p* are indicated in scatterplots. All *p* are two-sided, with no adjustments for multiple comparisons. Tukey boxplots used for discrete comparisons with Mann–Whitney U test. Centre line representing the median, box limits indicating the upper and lower quartiles, whiskers extending to 1.5 times the interquartile range.

but decreased co-stimulatory marker CD86 in cDCs, plasmacytoid DCs and classical monocytes (Supplementary Fig. 3d, f–h). Activation of Vδ2+ γδ T cells (CD86) and Tregs (HLA-DR and ICOS) was also higher in adults compared to children (Supplementary Fig. 3d, f, g, i).

Classical monocytes are key innate cell responders to malaria parasites[14] and are major contributors of the inflammatory cytokines associated with severe disease[17]. Additionally, monocytes are major contributors of the age-associated chemokines/cytokines present in the clinical data set (Fig. 1c), including CCL2[30], CCL3/CCL4[31] and IL10[32]. As such, we investigated if monocytes responded to *P. falciparum* parasites in an age-dependent manner. To investigate the impact of age on classical monocyte responses to malaria, PBMCs from malaria-naive children and adults were stimulated with *P. falciparum*-infected red blood cells (pRBCs) and production of IL10, IL6, IL1β, TNF and CCL2 quantified by intracellular staining (children $n = 13$, adults $n = 12$, Supplementary Fig. 4a). Classical monocytes were distinguished from non-classical monocytes based on CD64 expression (Supplementary Fig. 4b, c). CD64 has stable expression in culture, unlike CD16 which is down regulated[33], however intermediate monocytes may also be included within gated classical monocyte population. Stimulation with pRBCs increased the frequency of cytokine-producing classical monocytes compared to uninfected red blood cells (uRBCs) in both children and adults across all the cytokines tested (Fig. 2a). However, the proportion of CCL2+ classical monocytes was higher in adults, while children showed a significantly greater frequency of IL-10+ cells following stimulation (Fig. 2b). Further, amongst all cytokine-producing monocytes, the composition of parasite-induced cytokines differed significantly between adults and children. Although both children and adults had high frequencies of CCL2+ classical monocytes, children had a significantly more diverse cytokine profile, whereas adults had a CCL2-dominant response (Fig. 2c). Indeed, poly-functionality of classical monocytes was significantly lower in adults, who had a higher proportion of monocytes producing only a single cytokine (Fig. 2c, Supplementary Fig. 4d). Furthermore, there was a positive association between age and monocytes that produced only one cytokine, and a negative association between increasing age and monocytes that produced two or more cytokines suggesting monocyte polyfunctionality is age-dependent (Fig. 2d, Supplementary Fig. 4e). While age differences in the distribution of monocyte subsets have been reported previously[34] amongst our cohort there was no difference in the proportion of classical, non-classical, or intermediate monocytes between children and adults (Supplementary Fig. 4f). Together, these data show that classical monocyte responses to malaria parasites in naive individuals are influenced by age, with adults having a focused inflammatory response while, children had a poly-functional monocyte cytokine response with higher IL10+ frequencies.

## The transcriptional response to malaria parasites in classical monocytes is more inflammatory in naive adults

To explore the pathways driving the increased inflammatory response in monocytes from adults and increased immune regulatory response in children, we assessed the transcriptional profile of classical mono-cyte cells isolated ex vivo and following in vitro stimulation with *P. falciparum* malaria parasites (Supplementary Fig. 5a/b). Differentially expressed genes (DEGs) with age (children compared to adults), malaria stimulation (before compared to after stimulation), and genes which respond in an age-specific manner (significant for age and stimulation, and/or with a significant interaction between age and stimulation) were identified using glmmSeq[35], which fits a negative binomial mixed-effects model at the individual gene level. Using this approach, a total of 10,562 DEGs were identified (Fig. 3a, Supplementary Data 1). Malaria stimulation resulted in a large transcriptional change, with 10,395 DEGs identified. 26% of these (2735 DEGs) were also differentially expressed with age or had a significant interaction term between age and stimulation, indicating a large age-dependent

response to malaria in vitro (Fig. 3a, Supplementary Data 1). Principal Components Analysis (PCA) revealed differences between adults and children, as well as before and after parasite stimulation (Fig. 3b, Supplementary Fig. 5c). Age-dependent gene differences exemplified the polarization of response, with upregulated genes having a larger increase in adults, and down regulated genes having a larger decrease in children (Fig. 3c). DEGs were categorized into 14 groups, based on whether genes were higher or lower in children or adults prior to and after stimulation, and whether expression increased or decreased in response to parasites (Fig. 3d, Supplementary Fig. 6a/b). The largest group of genes (876 DEGs) were higher at baseline in children but increased in adults only in response to stimulation, while decreasing in children. These 876 DEGs had a significant term in the glmsseq model for either interaction and state (358), interaction and age (7), interaction only (500) or all three, interaction, state and age (11). The second largest group (619 DEGs) were genes that were higher in children at baseline, and increased in both children and adults, but with a larger increase in adults. These 619 DEGs had a significant term in the glmsseq model for either interaction and state (602) or all three (17). Both gene groups resulted in higher expression in adults after malaria parasite stimulation (Fig. 3d). Of all DEGs which were upregulated in response to stimulation in children or adults (1838), 35 had a significant term for interaction and age, 1028 for interaction and state, 137 for state and age, 559 for interaction only, and 79 for all three terms. Of the 1675 genes upregulated in adults following stimulation, 97% (1624) showed higher expression levels in adults compared to children post-stimulation. Conversely, among the 163 genes upregulated in children following stimulation, 91% (149) exhibited higher expression in children compared to adults post-stimulation (Fig. 3d). Interestingly, this differential response pattern occurs despite the majority of these genes (1567) having higher baseline expression in children than adults under ex vivo conditions. This is consistent with the higher baseline frequency of cytokine-expressing monocytes in children compared to adults in the uRBC control condition (Fig. 2a) and highlights that adult monocytes mounted a stronger transcriptional response to *Plasmodium* than children.

We performed Ingenuity Pathway Analysis (IPA) of all upregulated DEGs that were higher in adults following pRBC stimulation. Most enriched pathways were uniquely enriched in adults (Fig. 3e). Consistent with increased frequencies of classical monocyte producing inflammatory cytokines in adults, adult-specific upregulated pathways included inflammatory IFNγ signaling pathways, MHC Class II presentation, as well as chemokine signaling and the macrophage alternative activation signaling pathway (Fig. 3e). Furthermore, consistent with significantly higher frequencies of IL10 producing monocytes in children, the IL10 signaling pathway was upregulated in children. Further, Rho GDP-dissociation inhibitor (RHOGDI) signaling was also inhibited in adults but activated in children. This pathway refers to the regulatory mechanisms involved in modulating the activity of Rho GTPases which regulate inflammatory responses[36]. The top upregulated genes in adults after stimulation included the chemokine *CXCL11* and *SLAMF9* (Supplementary Fig. 6c). *CXCL11*, which was also higher ex vivo in adults (Supplementary Fig. 6c), has been shown to increase in circulation during malaria, particularly during a primary infection[37] and has roles in malaria pathogenesis[31]. *SLAMF9*, which was higher in adults after parasite stimulation, promotes the initiation of an inflammatory response in antigen-presenting cells such as monocytes[38]. Additionally, adults had higher expression of the MHC Class II gene *HLA-DRA* after stimulation, consistent with a more robust response with higher functional potential (Supplementary Fig. 6c). Consistent with IPA, upstream regulators predicted to be uniquely activated in adults were also indicative of an increased inflammatory response to malaria parasites (Fig. 3f). Transcriptional factors increased in adults compared to children included KLF6, which drives pro-inflammatory gene expression[39], and STAT4 which is a critical

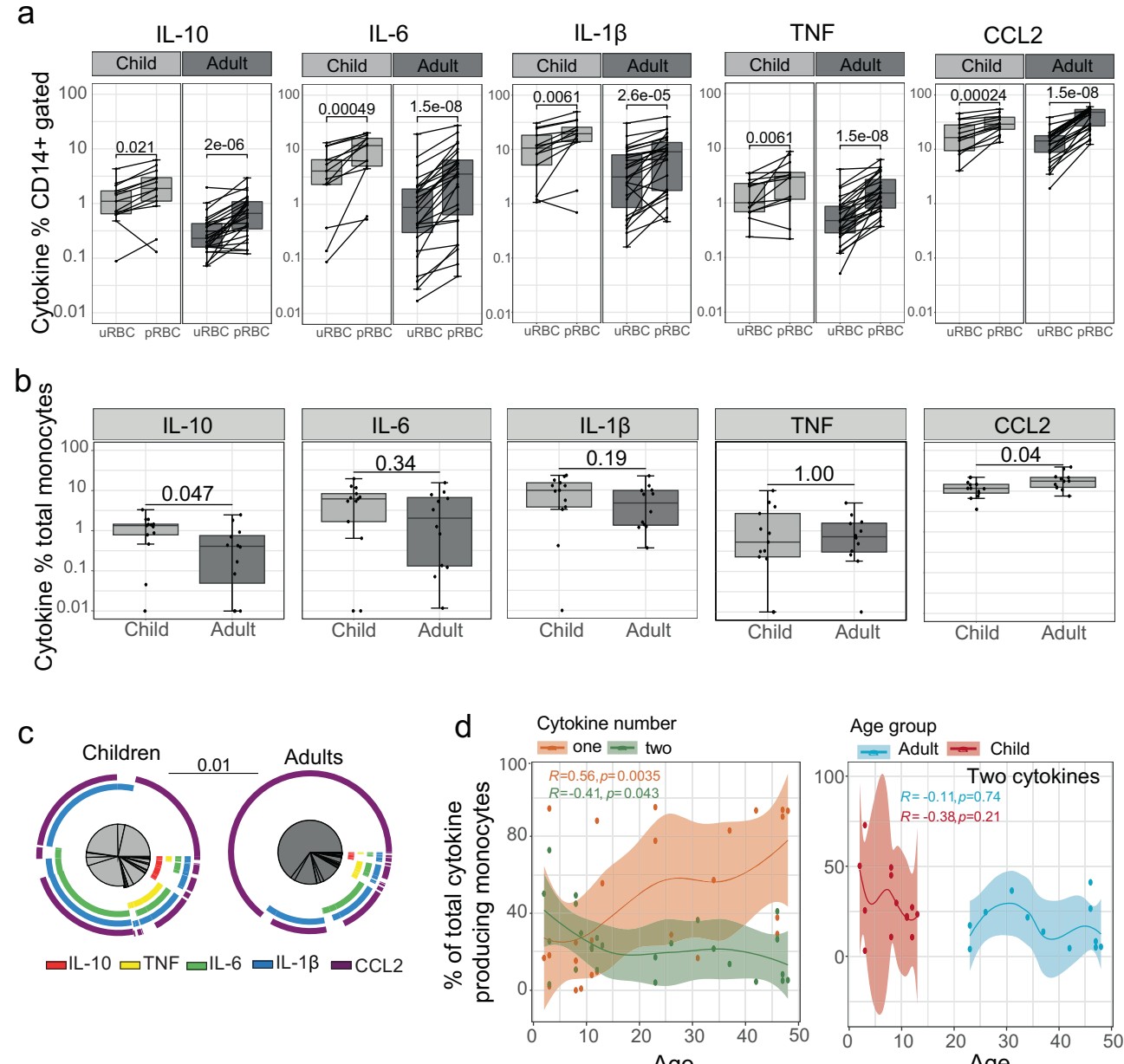

**Fig. 2 | Higher inflammatory cytokine production in response to *P. falciparum* in monocytes from malaria naive adults compared to children. a** Intracellular production of cytokines by classical monocytes were analysed following stimulation with *P. falciparum*·(*Pf*) infected (pRBCs) and uninfected red blood cells (uRBCs) with intracellular staining. **b** Proportion of *Pf* activated cytokine expressing classical monocytes (cytokine positive frequency in pRBC condition subtracted by uRBC condition). **c** Composition of cytokines from classical monocytes following parasite stimulation. **d** Left plot: correlation between number of cytokines produced by monocytes and age (years). One cytokine is orange, and two cytokines are green. Right plot: correlation between monocytes which make two cytokines and age. Adults are blue, children are red. Solid lines are LOESS fit curves with error bands of 95% confidence interval. Spearman's correlation and *p*-values are indicated. Lines represent paired observations, uRBC and pRBC comparisons are Wilcox rank paired test. Children and adult comparisons were made using the Mann–Whitney U test. Centre line representing the median, box limits indicating the upper and lower quartiles, whiskers extending to 1.5 times the interquartile range. Pie comparisons performed by Permutation test. Age associations were visualized by linear regressions and compared using Spearman's rank correlation. All *p* are two-sided, with no adjustments for multiple comparisons. For all panels data is from children *n* = 13 and adults *n* = 12. uRBC uninfected red blood cells, pRBC parasitised red blood cells, *Pf P. falciparum*.

driver of inflammation[40]. In contrast in children, upregulated transcriptional factors included HOXA3 which inhibits M1 polarization and drives M2 polarization[41], and KLF3, a suppressor of monocyte inflammation[42] (Fig. 3f). Inflammatory cytokines were predicted to be more activated in adults, including IFNγ, TNF, IL6, IL1β and type 1 IFNs (IFNα and IFNβ) (Fig. 3g). Consistent with an inflammatory response induced by parasites regardless of age, these cytokines were also identified as upregulated in children, albeit to lower levels (Fig. 3g). In

addition, CD40LG, IL-2 and CXCL12 were predicted to be activated in adults and inhibited in children. IL10 was predicted to be activated in both children and adults but was higher in children (Fig. 3g). Further, within the transmembrane receptor category of upstream regulators, Toll-Like Receptors 4 (TLR4), a pathogen recognition receptor that can bind to GPI anchors derived from the malaria parasite[43], was activated in adults and inhibited in children (Fig. 3h), consistent with an increased ability of monocytes from adults to bind parasite products

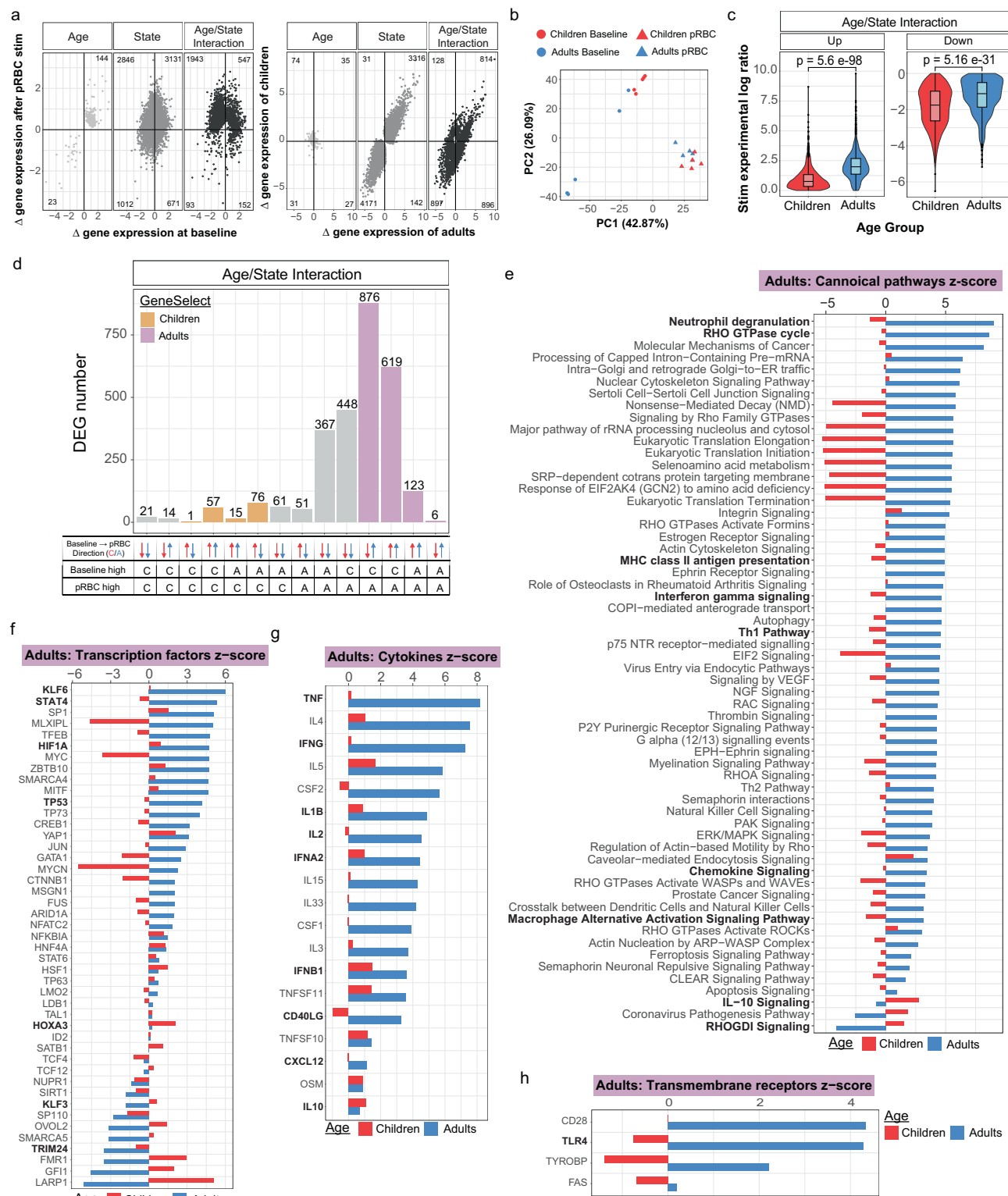

and induce inflammation. Analysis also identified upregulation of Th2 canonical pathways (Fig. 3e) and Th2 cytokines IL4/IL5 in adults (Fig. 3g). While Th2 responses are less studied in malaria, these data possibly indicate that adults can induce both Th1 and Th2 responses following parasite exposure or alternatively are due to overlapping genes in these pathways. In contrast, of the 149 DEGs (5%) which were higher in children after parasite stimulation (Fig. 3d), only two pathways were detected by IPA, with minimal differences in children and adults in upstream regulators (Supplementary Fig. 7a). Upregulated

cytokines identified in children nevertheless included TNF, IFNγ and IFNA, consistent with inflammatory responses induced by parasites regardless of age (Supplementary Fig. 7b–d). Taken together, transcriptional and cytokine production data show that classical monocytes' responsiveness to malaria is age dependent and suggests that there is a relatively enhanced inflammatory responses in adults compared to children in malaria-naive individuals. This finding is consistent with the major role of monocytes in malaria pathogenesis[14], the increased risk of severe disease in adults during primary infection[11],

**Fig. 3 | Increased transcriptional activation in response to *P. falciparum* in monocytes from malaria naive adults compared to children. a** Scatter plots describing the shape of these transcriptional data, differentially expressed genes (DEGs) grouped based on if they were significant for "Age", "State" or "Age/State Interaction". Left plots, show gene expression at baseline compared to after parasite infected red blood cells (pRBC) stimulation and right plots, show gene expression levels in malaria-naive adults (*n* = 5) compared to malaria-naïve children (*n* = 5). **b** Principal Components Analysis (PCA) of DEGs significant for age and state, at baseline and after parasite stimulation **c** Log ratio of genes separated by direction of change after pRBC stimulation. Centre line representing the median, box limits indicating the upper and lower quartiles, whiskers extending to the upper and lower limits. Storey's method was used for multiple adjustment of *p*-values **d** DEGs grouped based on their direction of change and relative expression levels

maximum in children (C) or adults (A) at baseline and after pRBC stimulation. Purple bars indicate genes that increased expression levels after stimulation and were higher in adults. Orange bars indicate genes that increased expression levels after stimulation and were higher in children. **e** Top 60 significant pathways identified by Ingenuity pathway analysis using the log ratio value for children or adults and the FDR adjusted *q*-value of the DEGs that were upregulated following stimulation and were higher in children (orange bars, **d**). Predicted upstream **f** transcription factors, **g** cytokines and **h** transmembrane receptors to be activated or inhibited in adults and children. Benjamin–Hochberg corrected *P*-values used to identify significant pathways and upstream regulators in the IPA analysis. All *p* are two-sided. For all panels data is from children *n* = 5 and adult *n* = 5. uRBC uninfected red blood cells, pRBC parasitised red blood cells, DEG Differential Gene Expression.

and the age associated levels of monocyte associated inflammatory cytokines CCL2, CCL3, CXCL8, CXCL9 detected in clinical malaria (Fig. 1). Nevertheless, analysis here reveal a diversity of DEG pathways, and specific experimental validation of these changes is required in future studies to directly link transcriptional changes to functional responses.

### Inflammatory cytokine production from Vδ2⁺ γδ T cells following malaria stimulation is higher in naive adults

Inflammatory transcriptional signatures in monocytes included the chemokine CXCL11, which has been linked to expansion of Vδ2⁺ γδ T cells in adults during a primary *P. falciparum* infection[37]. Vδ2⁺ γδ T cell, together with monocytes, are major producers of inflammatory cytokines associated with disease severity in malaria[17]. Following stimulation with *P. falciparum* parasites in vitro, Vδ2⁺ γδ T cells from children and adults responded robustly, with increased frequencies of IFNγ⁺ and TNF⁺ cells (Fig. 4a, Supplementary Fig. 8a). There were significantly higher frequencies of IFNγ⁺ and TNF Vδ2⁺ γδ T cells in adults, including both single-producing and IFNγ⁺/TNF⁺ co-producing cells (Fig. 4b, c). Within children, the frequency of IFNγ⁺ and TNF⁺ cells following stimulation was significantly increased from 0 to 10 years of age, highlighting the age dependent increase in inflammatory Vδ2⁺ γδ T cells (Fig. 4d). NK cells are also important innate cell responders to malaria parasites[16]. However, in contrast to robust Vδ2⁺ γδ T cell responses in vitro, we did not detect increased frequencies of cytokine producing NK cells following parasite stimulation (Supplementary Fig. 8b, c), consistent with our previous findings in similar in vitro systems[25].

The memory subset distribution of Vδ2⁺ γδ T cells has been reported to change with age, with a reduction in naive (CD27⁺ CD45RA⁺) Vδ2⁺ γδ T cells with age[23,24]. Consistent with this, the proportion of effector memory Vδ2⁺ γδ T cells (EM; CD27⁻ CD45RA⁻) was significantly higher in adults compared to children (Fig. 4e). IFNγ⁺ and TNF⁺ Vδ2⁺ γδ T cells follow parasite simulation were primarily EM cells compared to other cell subsets and were higher in adults compared to children (naive [NAIVE]; central memory [CM]; CD27⁺ CD45RA⁻; and terminally differentiated effector memory [EMRA]; CD27⁻ CD45RA⁺) (Fig. 4f). This suggests that the increased response in adults may be due to age-dependent changes to Vδ2⁺ γδ T subset distribution. However, when the frequency of IFNγ and TNF-producing cells was quantified within each memory subset, both EM and EMRA Vδ2⁺ γδ T cells had significantly high frequencies of cytokine-producing cells in adults compared to children (Fig. 4g). As such, the increased cytokine response of Vδ2⁺ γδ T cells in adults is due to both the expansion of responsive effector memory cells within the Vδ2⁺ γδ T cell compartment and an increased capacity of responding cells to produce cytokine.

Together, these data indicate that parasite-induced increases in cytokine-producing monocytes and Vδ2 γδ T cell frequencies are age-dependent. To investigate if baseline characteristics of PBMCs, which

also differed between children and adults (Supplementary Fig. 2), were associated with the cytokine production, we assessed the correlations between cell frequencies and activation profiles at baseline with frequencies of cytokine producing monocytes and Vδ2⁺ γδ T cells following parasite stimulation (Supplementary Fig. 9). In children, parasite induced IL1β and CCL2 from classical monocytes was positively correlated with baseline frequencies of classical monocytes and NK cells, respectively (Supplementary Fig. 9). Further, IL10⁺ classical monocytes was also positively correlated with baseline activation (CD86) of pDCs and cDCs in children (Supplementary Fig. 9). In adults, the frequency of TNF⁺ Vδ2⁺ T cells was positively correlated with baseline CD16 expression in multiple cell types, including CD4 T cells, pDCs and cDCs, along with activation (HLA-DR) of non-classical (CD16+) monocytes (Supplementary Fig. 9). IFNγ⁺ and TNF⁺ Vδ2⁺ T cell frequencies was also positively correlated with baseline activation of NK cells (CD86 and ICOS expression) (Supplementary Fig. 9). While further studies are needed to determine whether the correlations identified reflect mechanistic relationships, data are consistent with reported links between baseline immune compositions and cell responsiveness infection and vaccination[44], including in malaria[45,46].

### Transcriptional response to malaria parasites in Vδ2⁺ γδ T cells is similar in naive children and adults

As for monocytes, we next examined the transcriptional profile of purified Vδ2⁺ γδ T cells ex vivo and following stimulation with parasites to identify pathways mediating increased inflammation in adults (Fig. 5a, Supplementary Data 2). As seen for monocytes, Vδ2⁺ γδ T cells had a large transcriptional response to parasite stimulation, with 8766 DEGs identified after stimulation, of which 35% (3062) were also significant for age, or had a significant interaction with age (Fig. 5a). However, in contrast to classical monocytes, PCA revealed fewer differences between children and adults both before and after stimulation in the Vδ2⁺ γδ T cells (Fig. 5b, Supplementary Fig. 10a). Furthermore, DEGs upregulated with stimulation had larger increases in children, and down regulated genes had larger decreases in adults (Fig. 5c). When DEGs were grouped based on expression before/after stimulation, and directional change, only 710 genes were increased and were higher in adults following stimulation compared to 2015 DEGs which increased with stimulation and were higher in children (Fig. 5d). However, when these two gene sets were analysed by IPA, largely similar pathways with comparable enrichment scores were identified (Fig. 5e, Supplementary Fig. 10b/c). Of pathways with higher enrichment in children, the majority of these were involved in translation and metabolism (Fig. 5f). Similar upstream regulators were predicted to be enriched in Vδ2⁺ γδ T cells between adults and children (Supplementary Fig. 10d–f). Interestingly, gene expression levels of IFNγ and TNF were significantly increased after parasite stimulation but were not different between age groups (Supplementary Data 2). The disconnect between transcriptional and functional data may be due to the differences in

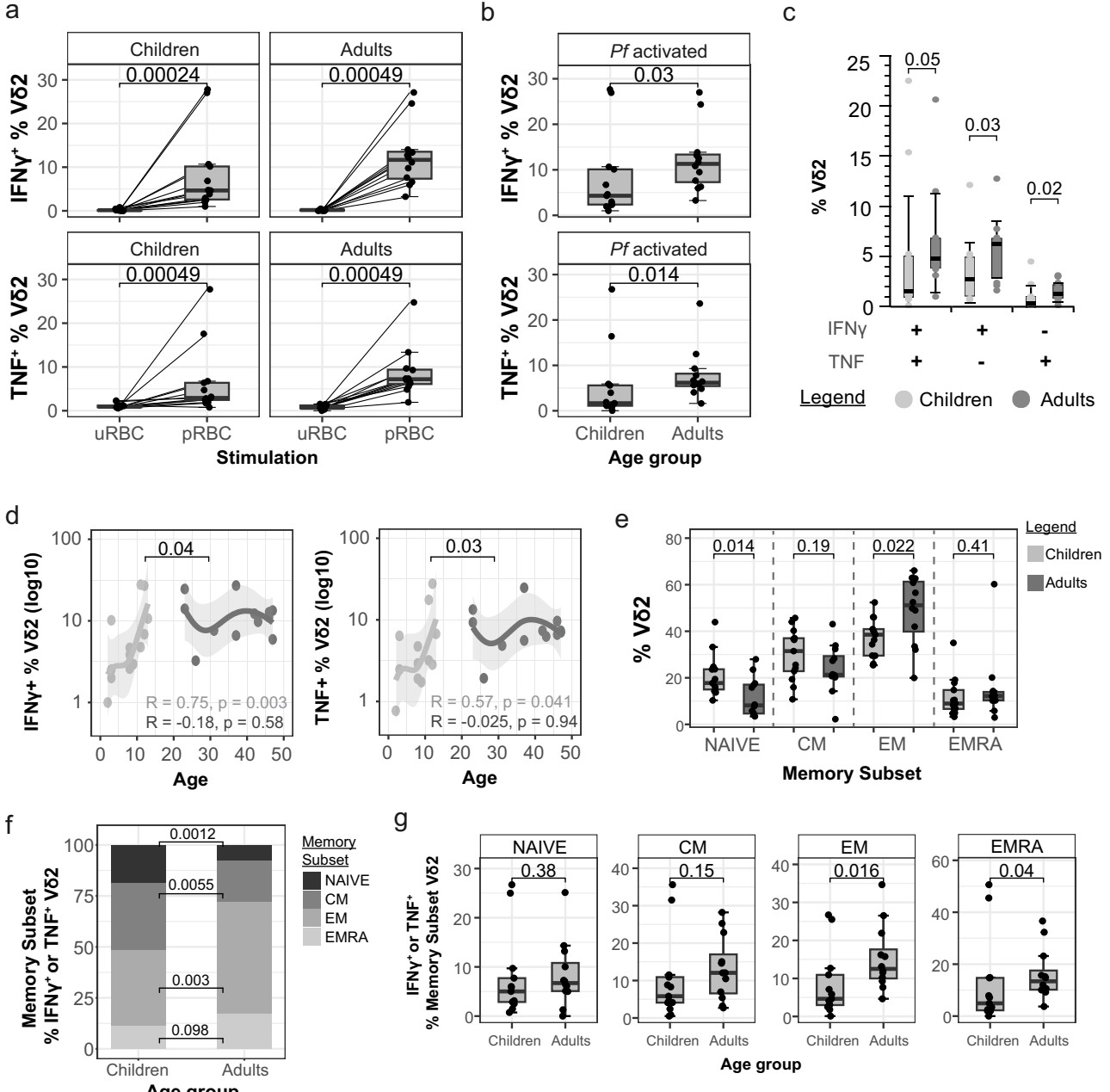

**Fig. 4 | Malaria-naive adult Vδ2⁺ γδ T cells produce more inflammatory cytokines following malaria stimulation. a** Intracellular production of IFNγ and TNF by Vδ2⁺ γδ T cells following 24-h stimulation with *P. falciparum* (*Pf*) infected red blood cells (pRBCs) and uninfected red blood cells (uRBCs) (children *n* = 13, adults *n* = 12). **b** Proportion of *Pf* activated cytokine expressing Vδ2⁺ γδ T cells (cytokine positive frequency in pRBC condition subtracted by uRBC condition). **c** Co-expression of cytokines by *Pf* activated Vδ2⁺ γδ T cells. **d** Proportion of Cytokine positive pRBC stimulated Vδ2⁺ γδ T cells and age in years. LOESS regressions and Spearman's rank correlation separated into age groups. Solid lines are LOESS fit curves with error bands of 95% confidence interval. **e** Vδ2⁺ γδ T cell memory subset

frequency comparisons after pRBC stimulation: NAIVE (CD27⁺CD45RA⁺), CM (CD27⁺CD45RA⁻), EM (CD27⁻CD45RA⁻) and EMRA (CD27⁻CD45RA⁺). **f** Memory subset proportions of cytokine producing Vδ2⁺ γδ T cells during pRBC stimulation. **g** Cytokine producing proportion of Vδ2⁺ γδ T cell memory subsets. Lines represent paired observations. Wilcoxon signed rank test used for paired data. Mann−Whitney U test used for unpaired data. Centre line representing the median, box limits indicating the upper and lower quartiles, whiskers extending to 1.5 times the interquartile range. All *p* are two-sided. For all panels data is from children *n* = 13 and adults *n* = 12. uRBC uninfected red blood cells, pRBC parasitised red blood cells, *Pf P. falciparum*, EM effector memory, CM central memory.

---

timing of assays, or due to post-translational age-dependent control of inflammation and IFNγ/TNF expression.

### Increased innate cell inflammatory responses in naive adults is not balanced by induction of FOXP3⁺ Tregs

Inflammation from innate immune cells in response to pathogens requires control to avoid immunopathogenesis. In malaria, control of

inflammation can be mediated by multiple cell responses, including Tregs[47]. Tregs expand during primary *Plasmodium* infection in adults, and expansion is associated with reduced inflammation[48]. Activation of Tregs has been linked to the immunoregulatory enzyme indoleamine 2,-dioxygenase (IDO)[49,50]. In our data plasma IDO was associated with age in clinical malaria (Fig. 1b/c) and was significantly upregulated in adults following parasite stimulation in the RNAseq monocyte data set

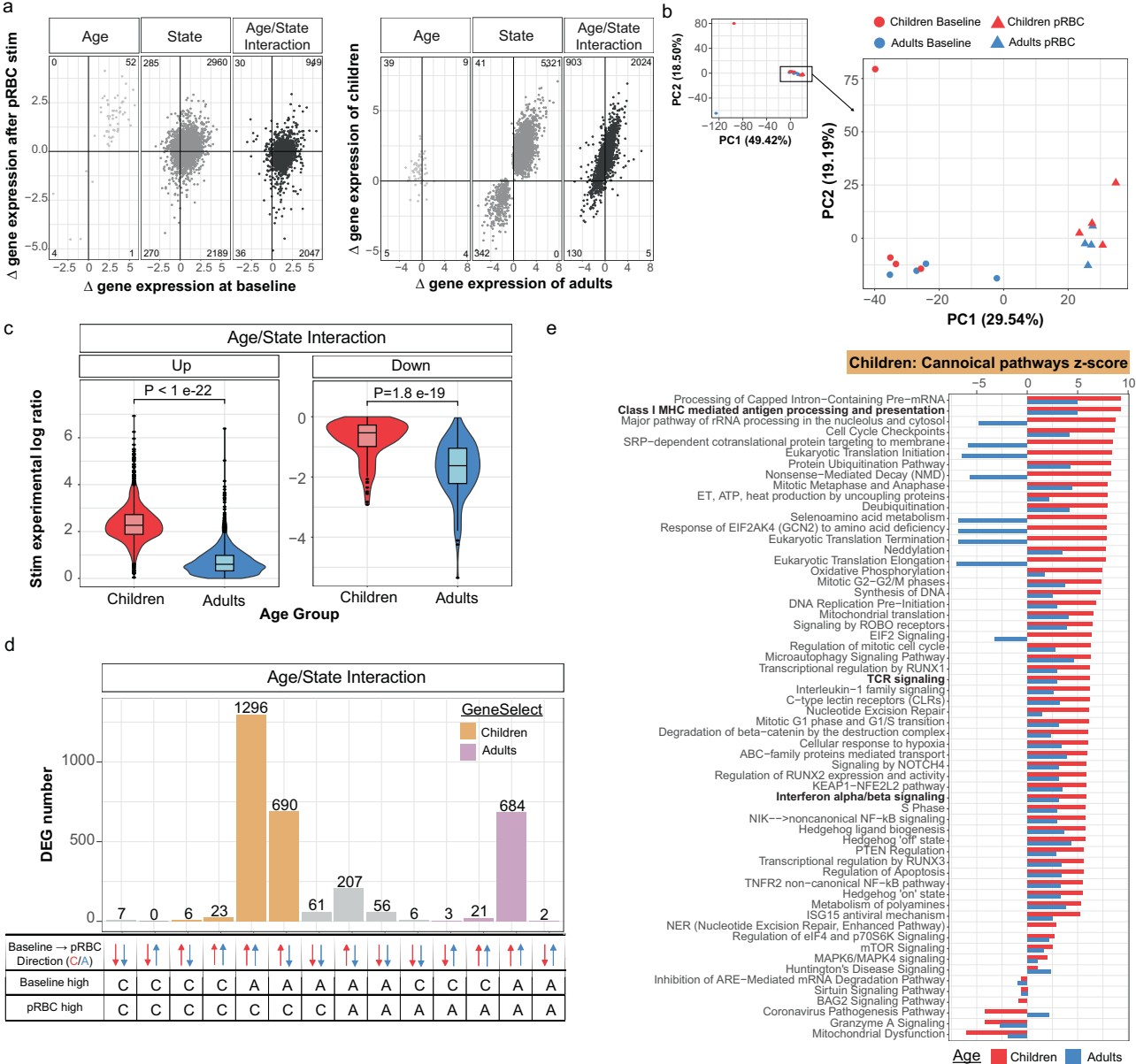

**Fig. 5 | Transcriptional activation of Vδ2⁺ γδ T cells by malaria is similar in naive children and adults. a** Scatter plots describing the shape of these transcriptional data, DEGs grouped based on if they were significant for "Age", "State" or "Age/State Interaction". Left plots, show gene expression at baseline compared to after pRBC stimulation and right plots, show gene expression levels in malaria-naive adults (n = 5) compared to malaria-naïve children (n = 5). **b** Principal Components Analysis (PCA) of DEGs significant for age and state, at baseline and after parasite stimulation **c** Log ratio of genes separated by direction of change after pRBC stimulation. Centre line representing the median, box limits indicating the upper and lower quartiles, whiskers extending to the upper and lower limits. R can only give an accurate P-value up to 22 digits hence, an estimate is given by R for 'up' genes. **d** DEGs grouped based on their direction of change and relative expression levels

maximum in children (C) or adults (A) at baseline and after pRBC stimulation. Purple bars indicate genes that increased expression levels after stimulation and were higher in adults. Orange bars indicate genes that increased expression levels after stimulation and were higher in children. **e** Top 60 significant pathways identified by Ingenuity pathway analysis using the log ratio value for children or adults and the FDR q-value of the DEGs that were upregulated following stimulation and were higher in children (orange bars, **d**). Benjamin–Hochberg corrected P-values used to identify significant pathways and upstream regulators in the IPA analysis. All p are two-sided. For all panels data is from children n = 5 and adult n = 5. uRBC uninfected red blood cells, pRBC parasitised red blood cells, DEG Differential Gene Expression.

(Supplementary Fig. 6d), suggesting that adults may induce higher Treg cell frequencies to control increased inflammation. Tregs can be expanded from malaria naive individuals following parasite stimulation in vitro[51–55]. As such, we tested whether age impacted the malaria induced expansion of Tregs in naive individuals, hypothesizing that increased inflammation in response to malaria parasites in adults would be balanced by increased induction of Tregs. Following stimulation, Tregs were identified as CD4⁺/CD25high/CD127low/Foxp3⁺ T cells, activation measured by CD38, ICOS or Ki67 expression and

inhibition potential measured by CCR4, TNFR2 and PD-1 expression[56] (Fig. 6a, Supplementary Fig. 11a/b). Treg expansion occurred to comparable levels in both children and adults (Fig. 6b). Parasite-expanded Treg populations had increased expression of activation markers compared to control-stimulated cells, but with no significant difference between age groups (Fig. 6c). Further, while the expression of inhibitory markers was higher in adults in cells stimulated with either parasite infected or uninfected RBCs (Supplementary Fig. 11c), there was no difference in the frequency of inhibitory marker expression

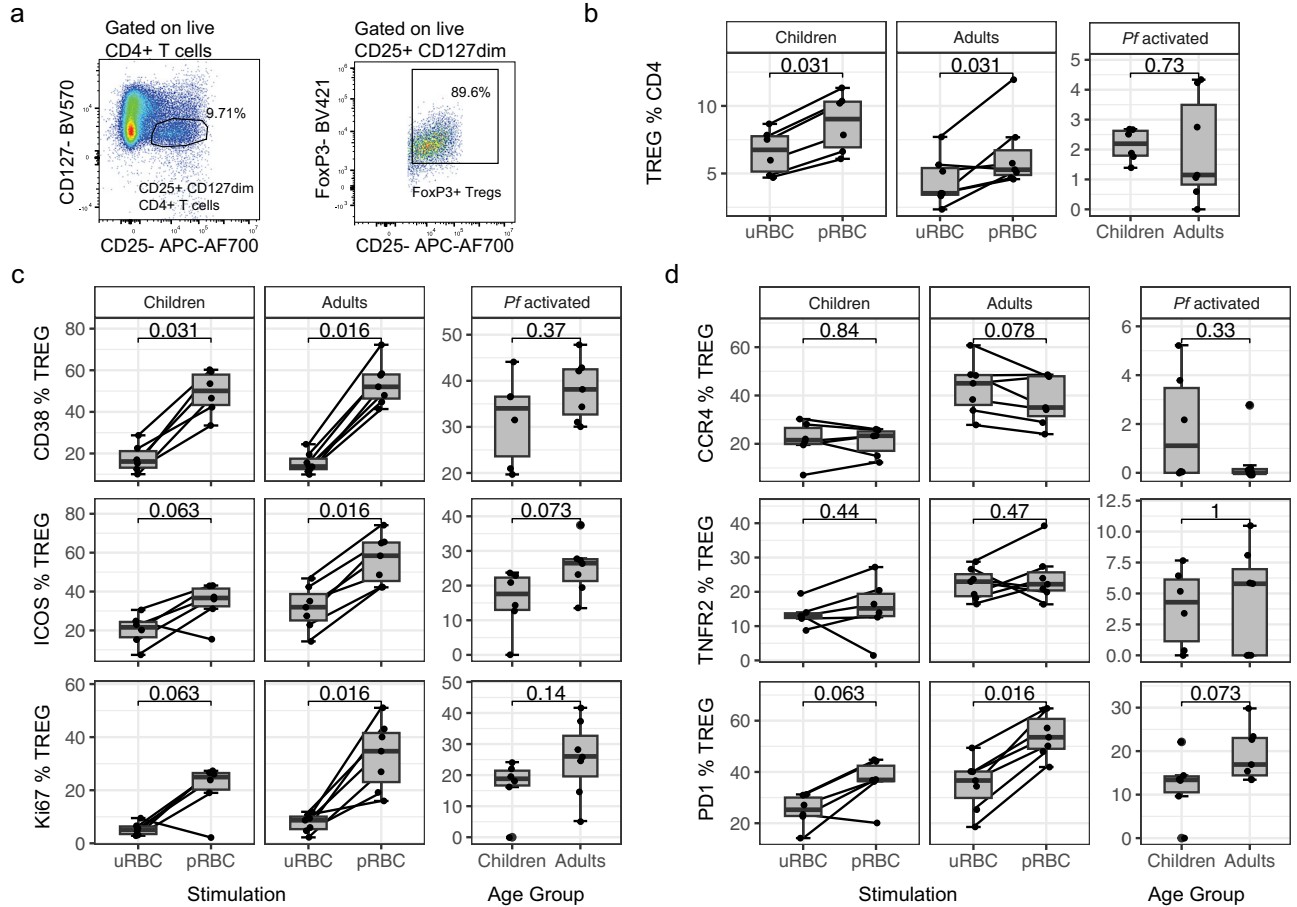

**Fig. 6 | Comparable Foxp3⁺ Treg response in both adults and children in response to malaria.** CD4⁺ T regulatory cell (Treg) frequency and surface marker expression in children (*n* = 6) and adults (*n* = 7) was measured after 5 days in vitro co-culture with trophozoite stage *P. falciparum* (*Pf*) red blood cells (pRBCs) and uninfected red blood cells (uRBCs). **a** Tregs were identified as CD25high CD127low and FoxP3⁺ CD4 T cells. **b** Treg frequency within CD4⁺ T cell compartment following stimulation. Surface marker frequency of inflammatory immune activation (**c**: CD38, ICOS and Ki67) and inhibition (**d**: CCR4, TNFR2 and PD−1) post

stimulation of Treg frequency with uRBC, pRBC and *Pf* activated responses (quantified as expression in pRBC minus expression in uRBC cultured conditions). Lines represent paired observations. Wilcoxon signed rank test used for paired data. Mann–Whitney U test used for unpaired data. Centre line representing the median, box limits indicating the upper and lower quartiles, whiskers extending to the upper and lower limits. All p are two-sided. For all panels data is from children *n* = 6 and adult *n* = 7. *Pf P. falciparum*, uRBC uninfected red blood cells, pRBC parasitised red blood cells, TREG FoxP3+ regulatory T cells.

between adults and children (Fig. 6d). Within these data, we also investigated whether the inflammatory CD4 T cell subset (Th1) and helper subset T-follicular helper (Tfh) CD4 T cells were expanded or activated during in vitro pRBC stimulation (Supplementary Fig. 12a/b). IFNγ production by CD4 T cells is associated with parasite clearance, but also disease[57], while Tfh cells play key roles in antibody development[58]. Similar to Tregs, parasite induced Th1 (CXCR3⁺ CCRR6−) CD4 T cells and circulating Tfh had increased expression of activation markers and inhibition markers compared to control-stimulated cells (Supplementary Fig. 12c–f). However, responses were largely comparable between children and adults for both CD4 T cells (Supplementary Fig. 12c/d) and Tfh cells (Supplementary Fig. 12g), suggesting that the increased inflammatory responses in monocytes and γδ T cells is not a global enhancement of inflammation with all cells of the immune response but rather specific to innate cells.

## Discussion

Here we identify an age-dependent inflammatory response to *P. falciparum* both during human disease and in vitro, that may contribute to age-dependence of malaria pathogenesis and disease severity risk[9,11,12]. In a cohort of patients with clinical malaria, plasma levels of inflammatory cytokines were associated with age and disease symptoms.

This is despite age being independent from circulating parasitemia. In unexposed individuals, both monocytes and Vδ2⁺ γδ T cells had significantly higher frequencies of cells producing inflammatory cytokines in adults compared to children following in vitro parasite stimulation. Age-dependent transcriptional activation signatures were also detected. This higher inflammatory potential was not balanced by more robust induction of Treg cells, nor was there age associated differences in Th1 or Tfh CD4 T cell responses. Together with previous studies in the same clinical population showing age-dependence of endothelial activation in *falciparum* malaria[9], our findings suggest that age-dependent immune-mediated inflammation contributes to age-dependent disease risk in malaria.

In the absence of migration of naive individuals into areas of disease risk[11], or emergence of new pathogens such as SARS-CoV-2, understanding age-intrinsic factors influencing the immune response to infection is confounded by prior exposure and the development of adaptive immunity. Thus, while it is well recognized that the immune system evolves with age, with important consequences for disease risk[1–3], few studies have defined age-specific mechanisms between children and adults in responses to individual pathogens. Here, we identify that both monocytes and Vδ2⁺ γδ T cells are significantly more inflammatory in adults compared to children in response to malaria.

Monocytes and Vδ2[+] γδ T cells have been shown to be major producers of cytokines associated with severe malaria[17]. Monocytes produce inflammatory cytokines to malaria parasites via several pathways[14], including via TLR2 and TLR4 recognition of Glycosylphosphatidylinositol (GPI) anchors[43], TLR8 responding to *Plasmodium* RNA[59], TLR9 responding to hemozoin and *Plasmodium* DNA[60], and via TLR4 binding hemozoin bound with parasite proteins[61]. Previous studies have shown that monocyte responsiveness to TLR2, TLR4 and TLR9 stimulation varies with age, but responsiveness to TLR7/8 is age-independent[62–65]. This suggests that the age-dependent response to malaria may be driven by specific pattern recognition pathways. Indeed, data presented here suggests that increased frequencies of cells producing inflammatory cytokines in response to malaria parasites may be mediated by increased TLR4 activation, which was transcriptionally activated in monocytes in adults and not children in response to parasite stimulation.

Amongst the top upregulated genes in monocytes from adults following parasite stimulation was the chemokine CXCL11, which acts as a chemotactic for activated T cells and is increased in symptomatic malaria compared to asymptomatic infected individuals[66]. CXCL11, along with CXCL9 has also been linked to expansion of Vδ2[+] γδ T cells in adults during a primary *P. falciparum* infection[37]. Consistent with this link, Vδ2[+] γδ T cells from malaria-naive adults have higher frequencies of IFNγ and TNF-producing cells in response to malaria in vitro stimulation compared to children and circulating CXCL9 was correlated with age in patients with malaria. This increased inflammatory response was both due to the expansion of EM cells within the Vδ2[+] γδ T cell population and increased reactivity of all memory subsets to parasite stimulation. Surprisingly, we did not identify transcriptional pathways that were differentially activated that could explain these age-dependent cytokine profiles. While this may suggest that cytokines are instead regulated post-translationally, differences in timing of stimulation should also be considered (24 h stimulation for intracellular cytokine quantification, compared to 4 h for RNAseq analysis). Along with roles of Vδ2[+] γδ T cells during infection, these cells also have been associated with protection induced by whole parasite irradiated sporozoite vaccines[67,68]. While these vaccines induce sterile protection in malaria-naive adults, they have failed to induce any efficacy in infants in endemic areas with this failure hypothesized to be linked to Vδ2[+] γδ T cell function[69]. Our data support this hypothesis by identifying a clear age-dependent responsiveness of γδ T cells to malaria parasites.

In contrast to responses in adults, monocyte cell responses in children had increased regulatory potential, with both transcriptional enrichment of regulatory pathways including IL10 and RHODGI signaling in children, as well as increased frequencies of IL10 producing cells in response to parasite stimulation. In malaria endemic areas, responses from monocytes isolated from immune Malian adults produce more IL10 and less inflammatory cytokines (TNF, IL6 and IL-1β) compared to both non-immune children and malaria-unexposed adults[19], consistent with the induction of immunoregulatory networks by malaria exposure associated with protection from disease[18]. Here we show that in naive populations, monocytes from children also produce more IL10 compared to adults, in an age-intrinsic manner, distinct from malaria-induced regulator networks. In contrast, while frequencies and functions of Tregs cells are highest early in life[70], we found no evidence that induction of Tregs by malaria parasites was age dependent. However, it should be noted that circulating IDO levels were increased with age in patients with malaria, so increased Treg induction may still occur during clinical disease. Further studies are required to investigate the age-dependent induction of other regulatory cell responses during malaria, such as Tr1 CD4 T cells which co-produce IL10 with IFNγ and dominate the CD4 T cell compartment during malaria infection[71,72].

While the mechanisms underpinning the age-dependent changes to monocytes and Vδ2[+] γδ T cell functions in response to parasites observed here remain to be elucidated, one important possibility is 'trained' immunity in both monocytes and Vδ2[+] γδ T cells. Trained immunity in innate cells is the long-term functional reprogramming of responses induced by a specific stimulation, that results in an altered and increased response to a secondary alternative stimulus[73]. For monocytes, training has been reported following a diversity of stimulations including Bacillus Calmette-Guérin (BCG) vaccination[74], β-glucan[75] and influenza vaccination[76] and can be driven by reprograming of the haematopoetic stem cell compartment, thus transmitting sustained changes to monocytes. BCG vaccination has also been reported to induce innate immune memory in γδ T cells[77]. The direct impact of BCG trained immunity on malaria responses has been tested in a small controlled human malaria infection study of 20 individuals, with results suggesting that BCG induced trained immunity increased inflammatory responses and diseases severity[78]. Malaria-naive participants in our study, including adults, are unlikely to have received BCG vaccination which is not routinely administered in Australia. However, the history of other pathogens and vaccinations in adults compared to children could have resulted in modified 'training' that underpins different inflammatory responses to malaria parasites. Indeed, influenza vaccination drove increased responses to unrelated viruses Zika and Dengue[76], and can modify responses to COVID-19[79], providing proof of concept that routine vaccinations can modify responses to unrelated pathogens. Within our malaria naïve cohorts, we also detected correlations between baseline cell frequencies and activation profiles and parasite-induced cytokine production. Studies focused on whole blood transcriptomic profiling have shown that baseline signatures can predict antibody responses across multiple vaccines, with non-classical monocytes and cDCs the major cell types driving these predictions[80]. These key cell types were also associated with inflammatory cytokine production following parasite stimulation in our study. Together, data highlight the importance of baseline immune profiles and immune responses to pathogens[44].

Our data are of particular relevance for areas where malaria transmission is decreasing, including in populations where infants and young children are protected by World Health Organization recommended RTS,S or R21 vaccination. In both cases, first infections may occur in older individuals, or following waning of vaccine-induced protection. However, while increased inflammatory responses with age may increase disease severity and risk, this heightened responsiveness may also be beneficial for the development of adaptive immunity in response to infection. In malaria naive children and adults moving into high endemic areas, while adults had initially increased risk of severe disease, they acquired protection from infection more rapidly[11,81]. This age-dependent acquisition of anti-parasitic immunity, largely mediated via adaptive responses, has also been shown in Ugandan children in high transmission areas[82]. For children protected from malaria via vaccination, further studies are required to understand the severity of 'rebound' infections and immune development to infection after waning of vaccine efficacy. However, clinical trials of infants who received dihyrdoartemisinin-peperaquine to prevent malaria between 8 weeks to 24 months of age, showed that children who received 4 weekly doses (but not 12 weekly doses) of chemoprevention also had reduced episodes of malaria in the year following drug cessation[83]. While the mechanisms of this prolonged protection are unknown, the chance for infant immune system maturation prior to infection may be a factor. Indeed, while we did not detect differences in CD4 T cell activation in vitro in this study, we have previously linked increased antibody development after malaria in adults to increased activation of T-follicular helper cells during malaria[84]. Further studies are required to investigate innate and adaptive responses during malaria across age. Additionally, studies have shown that innate cell responsiveness in children differs profoundly across geographic

location, with marked lower functional responses in South African compared to Belgian or Canadian children[85]. As such, how our findings translate to clinical disease and immunity in children in areas of high malaria transmission, who also experience high burdens of other pathogens and unique environmental exposures, requires further investigation.

Limitations of our study include the absence of both young infants and the elderly in both our unexposed healthy and natural infection cohorts. Rapid maturation of the immune system in infants may further modulate the age responsiveness to malaria reported here and would likely show differences to the children studied here. Indeed, changes to TLR stimulation in myeloid cells rapidly adapt in the first year of life[64], and γδ T cells gain effector functions in the first months after birth[22]. Further, risk of severe disease also increases in older adults, compared to young adults[9,12,86], thus future studies are required to understand innate cell responsiveness to malaria parasites in these age groups. Due to the unequal risk of malaria between sexes in malaria cohorts[26,27], we were also unable to explore sex related differences on inflammatory responses during malaria. Indeed, sex is an important factor which should be considered in future studies, particularly considering previously reported differences in TLR responsiveness[87], sex modulated malaria induction of FoxP3+ Tregs *in utero*[88], and sex-based differences in clearance of asymptomatic parasite infection[89]. Limited dynamic range of Luminex approach to assess inflammatory cytokines resulted in a disproportionate number of children with values below the lower limit of detection. While we cannot explore the age association of these analytes below the detection limit, data are indicative of an overall lower inflammatory response in children. More sensitive approaches, such as proximity extension and proximity ligation assays such as Olink or Alamar NULISAseq[90], would allow a better understanding of age-dependent inflammatory changes in children and a more comprehensive analysis of analytes which were not detectable by Luminex. Additionally, no cellular samples were available from our clinical cohort of natural infection, thus we were not able to investigate age-dependent changes to monocytes and γδ T cell activation during clinical malaria. Further, due to limited cell numbers in malaria naive donors, we were also unable to investigate the specific innate cell pathways linked to differences seen following parasite simulation (for example, TLR2 or TLR4 pathways). Additionally, as this cohort was recruited from an allergy clinic, due to the difficulties in collecting blood from children without a clinical need, we are unable to formally exclude that underlying presence of allergies may impact findings. We are unable to differentiate between differences induced by cell intrinsic compared to cell-extrinsic responses. Indeed, cell composition changes dramatically with age, and these changes may contribute to differences in monocyte and γδ T cells cytokine production. Further studies are required to identify cell intrinsic and extrinsic factors and other innate cell responses which may have roles in malaria disease risk. Indeed, early responses to parasites both in vitro and during primary malaria infection robustly activate DCs[91], which were not specifically investigated in our study, but no doubt play pivotal roles in immune responsiveness. Furthermore, we assessed malaria parasite immune responses with non-viable parasites from cryopreservation. While this approach allowed for a single batch of parasites to be used across all assays in this study, different responses may have been detected when using viable parasites and this should be assessed in future studies.

In conclusion, these data identify classical monocytes and Vδ2+ γδ T cells as important drivers of the inflammatory response in malaria-naive adults upon first exposure to malaria parasites. Age-dependent inflammatory responses and similar age-dependence in endothelial activation[9], may contribute to the increased risk of severe disease in adults compared to children in low malaria transmission settings. These findings have important implications for our understanding of the distinct immune response in children and adults upon first exposure to a pathogen. In addition to informing age dependence of inflammation in malaria, our findings may be of relevance to other infectious diseases where severity follows 'J' or 'U' shape age distributions[2,3].

## Methods

### Study participants

**Clinical cohorts.** Age related circulating cytokines/chemokines and clinical associations were analyzed in all available participants from patients ($n = 76$) enrolled in previously published studies of patients reporting with *P. falciparum* malaria at district hospitals in Kudat Division, northwest Sabah, Malaysia[24,26], and in an additional 18 individuals enrolled at a referral hospital for the West Coast and Kudat Division[25,27] (Supplementary Table S1). Patients presenting to study hospitals with microscopy-diagnosed malaria of any species were enrolled following written informed consent. Children were pre-defined as age <12 years, based on Malaysian Ministry of Health pediatric ward admission guidelines. Sex was assigned by enrolling study staff based on identity card and/or hospital records and information on gender was not collected. For all cohorts, a higher proportion of infected males compared to females was observed for all malaria species, possibly because of infection risk of forest workers[26,27]. For the current analysis, patients were included if *Plasmodium* species PCR-confirmed *P. falciparum* mono-infection. Laboratory analysis for clinical parameters was conducted as previously described in parent studies. Clinical data from the district hospital cohort[26] was reanalyzed based on age with children defined as <12 years, compared to adults >12 years (Supplementary Table S2). Here, plasma samples collected from lithium heparin blood collection tubes during enrollment were used. In this study area, we have previously reported 100% CMV sero-positivity in a cohort of 43 individuals[92].

**Malaria-naive children and adults.** Blood samples were collected from children ($n = 13$, 61% male, age 8 [3–12] years (median [IQR])) and adults ($n = 13$, 46% male, age 42 [29–46] years (median [IQR])) from an outpatient allergen clinic at the Royal Darwin Hospital, during blood collection for routine care (Supplementary Table S3). Patients were attending the clinic for specialist appointments for routine allergy management and follow-up but were currently free of all allergy symptoms and were otherwise well. Inclusion criteria was diagnosis with food or other allergy but allergic symptom free for at least 2 weeks (children aged 2–12, adults aged 18–59). Exclusion criteria were use of immune-modulatory medication in the past 2 months, active eczema or asthma and any other illnesses. This recruitment strategy allowed for the collection of blood from children who were otherwise healthy. Allergies for children and adults were similar, both groups had participants with nut, dust mite, and food allergies (Supplementary Table S3). The majority of participants had not experienced an allergic reaction for >6 months, were malaria-naive and were confirmed healthy by an on-site immunologist. Peripheral blood mononuclear cells (PBMCs) were separated by ficoll-paque density gradient centrifugation and cryopreserved. As expected, children had a higher absolute lymphocyte count (Children: 2.8 [2.5–3.7] x10$^9$/µL (median [IQR]); Adults: 2.1 [1.7–2.5] x10$^9$/µL (median [IQR]), $p = 0.002$), while there was no difference in the whole blood monocyte count (Children: 0.5 [0.5–0.6] x10$^9$/µL (median [IQR]); Adults: 0.5 [0.4–0.6] x10$^9$/µL (median [IQR])). Sex was self-reported or taken from medical records. Gender was not recorded. CMV serostatus was assessed by commercially available ELISA kits (ab108724) according to manufacturer's instructions. CMV seropositivity did not differ between children and adults (Supplementary Table S3).

### Multiplex assay

ProcartaPlex human custom kits were purchased from Thermo Fisher Scientific (MA, USA) focusing on inflammatory cytokines and analytes

with roles in malaria pathogenesis and run according to manufacturer's recommendations with the following modifications[90]. Cryopreserved plasma from individuals with naturally acquired malaria during acute infection were diluted 3X before performing assay with a custom kit (Cat #: PPX-27-MXNKUV6) and 30,000X for human simplex CRP kits (Cat #: EPX01A-102 88-901). After sample dilution, 25 µL of plasma was incubated with 25 µL of 1X Universal Assay Buffer (Cat #: EPX-11111-000) and magnetic capture beads at 4 °C overnight on an orbital shaker set to 600 rpm. Plates were washed twice then incubated with Biotinylated detection antibody at RT for 30 min at 600 rpm. Plates were washed twice then incubated with Streptavidin-PE at RT for 30 min at 600 rpm. Plates were washed twice, resuspended in Reading Buffer then acquired on the Bio-Rad Bio-Plex 200 array reader. Wells with a bead count of ≥50 were included in analysis. Optimised standard curve 4PL/5PL regressions were calculated for each analyte based on analyte specific standards with Bio-Plex Manager software and used to convert bead fluorescence intensity minus background to concentration (pg/mL) dependent on the standard concentrations provided by the manufacturers. Analytes with a quantification rate lower than 20% across all samples were removed from subsequent analysis (ANG1, IFNα, IFNγ, IL1β, IL12, IL6, IL4, TNF, RANKL, TREM1). Analytes with batch differences ($p \leq 0.05$, Mann–Whitney U test) in the same samples were median-ratio normalised across batches. Sample analyte values beyond the ULOQ or below the LLOQ were set to ULOQ +1 and LLOQ −1 (minimum = 0.001), respectively. ULOQ +1 and LLOQ −1 values were standardized across batches.

### Parasite culture

Packed red blood cells (RBCs) from donors were infected in vitro with the *Plasmodium falciparum* 3D7 parasite strain[93]. Packed RBCs for parasite culture were acquired from the Australian Red Cross. *Plasmodium falciparum* infected RBCs (pRBCs) were cultured at 5% haematocrit in Roswell Park Memorial Institute 1640 media (RPMI) supplemented with AlbuMAX II (0.25%) and heat-inactivated human sera (5%). Cultures were incubated at 37 °C in 1% $O_2$, 5% $CO_2$, 94% $N_2$ gas mixture. Culture media was replaced daily, and parasite stage/parasitemia was monitored by Giemsa-stained blood smears. pRBCs were grown to 15% parasitemia and purified from uninfected RBCs (uRBCs) and early stage pRBCs, via magnet separation to enrich mature trophozoite stage pRBCs. Purified pRBCs (>95% purity) were stored at −80 °C following addition of a Glycerolyte cryopreservant.

### Flow cytometry

**Ex vivo PBMC phenotyping.** Ex vivo PBMC phenotype and activation were assessed by flow cytometry. PBMCs were thawed in 10% FBS/RPMI, and surface staining performed in 2% FBS/PBS using fluorescent-tagged antibodies to identify cell lineages and measure activation marker expression of interest (Supplementary Fig. 2, Supplementary Table S4).

**Innate cell pRBC stimulation assay.** Parasites used in stimulation were non-viable, thawed from glycerolyte preserved trophozoite stages. Cryopreserved parasites were chosen so that one single batch of parasites could be used for all assays in this study. Glycerolyte preserved parasites were thawed by dropwise addition of malaria thawing solution (MTS: 0.6% NaCl in $H_2O$) and incubated at RT for 10 min. Parasites were washed in MTS/PBS (phosphate bufferd saline) three times with decreasing solution ratios (1:0, 1:1, 0:1). Following thaw, parasites were intact late stage trophozoites. PBMCs were co-cultured at 37 °C, 5% $CO_2$ in a 96-well U-bottom plate for 24 h at a 1:1 pRBC:PBMC ratio with $1 \times 10^6$ mature trophozoite stage pRBCs or $1 \times 10^6$ uRBCs, based on cytokine activation in optimization experiments (Supplementary Fig. 13)[20,21]. A 1:1 ratio of parasites to PBMCs is equivalent to approximately 4 log10 parasites/µl, based on 600:1

RBC:WBC ratio in adults and ~$6 \times 10^6$ RBC/µl. Thus, is equivalent to parasitemia seen in clinical cohort used here. Protein transport inhibitors (Monensin, BD GolgiStop) were added after 1 h at 37 °C, 5% $CO_2$. At 24 h, cells were stained to identify monocytes and Vδ2 T cells (Supplementary Table S5), washed with 2% FBS/PBS, cells were permeabilised with Cytofix/Cytoperm Fixation (BD Biosciences: Cat #554714) and stained for intracellular cytokine production (Supplementary Table S5). To determine the pRBC specific response uRBC responses were subtracted from pRBC responses.

**Treg/ CD4 T cell expansion assay.** To measure in vitro CD4 and Treg activation and expansion we performed a 5 day PBMC co-culture with pRBC at a 3:1 cell ratio, as performed previously[94]. A 3:1 ratio of parasites to PBMCs is equivalent to approximately 4.4 log10 parasites/µl, based on 600:1 RBC:WBC ratio in adults and ~$6 \times 10^6$ RBC/µl. Thus, is equivalent to parasitemia seen in clinical cohort used here. After 5 days, cells were washed with 2% FBS/PBS and surface stained to identify Tregs and Treg activation (Supplementary Table S6). To determine the pRBC specific response uRBC responses were subtracted from pRBC responses. FACS data was acquired using the Cytek Aurora 3 laser (CA, USA), and analysed with FlowJo v10 (BD, 2019).

### Cell isolation and RNA sequencing

PBMCs were thawed in 10% FBS/RPMI. Classical monocytes and Vδ2 T cells subsets were FACS sorted using the BD FACSAria III Cell Sorter from ex vivo PBMC or after in vitro co-culture at 37 °C, 5% $CO_2$ in 96-well U-bottom plates for 4 h at a 1:1 cell ratio with $1 \times 10^6$ mature trophozoite stage pRBCs (Supplementary Table S7). RNA was extracted from isolated cell population lysates using the QIAGEN PicoPure RNA isolation kit (Applied Biosystems, KIT0204), and RNA quality confirmed with the 2200 TapeStation system (G2964AA) by High Sensitivity RNA ScreenTape (5067- 5579). RNA sequencing libraries were constructed using the NEBNext Single Cell/Low Input RNA Library Prep Kit for Illumina® (E6420S) and NEBNext Multiplex Oligos for Illumina (96 Unique Dual Index Primer Pairs) (E6440S). The libraries were sequenced using a paired-end NextSeq 500/550 high output kit v2.5 (150 cycles) (Cat number 20024907). Individuals used for RNA-seq analysis were sex matched (children: 60% male; adults: 60% male).

### Data analysis

**Flow cytometry.** For the ex vivo phenotyping panel we used the R package SPECTRE[95] to identify cell subsets of interest. Unsupervised clustering was performed using both children ($n = 12$) and adult ($n = 11$) PBMC sample, and cell clusters were visualised with uniform manifold approximation and projection (Supplementary Fig. 2a/b, Supplementary Table S3). Expression of lineage markers were used to annotate cell clusters into 12 high level cell states (Supplementary Fig. 2a/b). For functional assays, data was analysed with FlowJo v10 (BD, 2019) and R/ R studio was used for statistical analysis and data visualisation.

**Bulk-RNAseq.** Raw sequencing reads were first trimmed to remove adapter sequences and low-quality bases using Cutadapt (v1.9). Trimmed reads were then aligned to the Human GRCh37 reference genome, incorporating Ensembl v97 gene models, using STAR (v2.5.2a). Alignment files were processed, sorted, and converted to the required formats using SAMtools (v1.9). Gene and transcript expression levels were quantified with RSEM (v1.2.30), providing normalized expression estimates. Quality assessment of the RNA-seq data was performed using RNA-SeQC (v1.1.8) to ensure data reliability. All analyses were carried out using Python (v3.6.1) and Perl (v5.22) for scripting and workflow automation. Low-count genes (fewer than 10 counts) were removed, and dispersion was estimated using edgeR (v4.40) workflow in R. A negative binomial distribution via regression models of normalized count data and Wald test was used to compare gene expression variation between paired pre- and

post-stimulated samples from children and adults with the glmmSeq R package[35]. The design matrix accounted for random effects of individual samples (Supplementary Fig. 4b) and included an interaction term between state (pre- vs. post-stimulation) and age (adult vs. child). Fold change due to stimulation was calculated by subtracting the log-transformed response term of unstimulated from stimulated samples. Fold change due to age was calculated by subtracting the log-transformed response term of children from adults for both stimulated and unstimulated samples. We corrected for multiple testing using Storey's $q$-value method, defining significance for $q$-values lower than 0.05 with the $q$-value R package.

IPA (Qiagen, Hilden, Germany) was used for canonical pathway enrichment and predicted upstream regulator analysis using DEGs with a false detection rate (FDR) < 0.05, with no fold-change cut-off.

### Statistical analysis

Non-parametric testing was performed for all analysis. Continuous data for all cellular responses and plasma cytokines were compared between the children and adults' groups using Mann–Whitney U test or correlated with age as a continuous variable with Spearman's correlations.

Statistical comparisons were not adjusted for multiple comparisons unless indicated. All analyses were performed in R (version 4.4.4). Graphical outputs were made in ggplot2 (version 3.5.1) and ggpubr (version 0.6.0). No sample size calculation was performed, instead all available participant data was included. No subgroup analysis was performed. Data generation was performed with blinding to participant demographic data.

### Ethics

Written informed consent was obtained from all study participants or, in the case of children, parents or guardians. Studies were approved by the ethics committees of the Northern Territory Department of Health and Menzies School of Health Research (Darwin, Australia, HREC 2010-1431, HREC-2012-1766), Medical Research and Ethics Committee, Ministry of Health, Malaysia (NMRR-10-754-6684 and NMRR-12-499-1203), QIMR-Berghofer Human Research Ethics Committee (HREC P3445 and P3444) and the Alfred Hospital Ethics Committee (HREC 188/23 and 80/24).

### Reporting summary

Further information on research design is available in the Nature Portfolio Reporting Summary linked to this article.

## Data availability

The fastq files and raw counts for transcriptional data generated in this study have been deposited in Gene Expression Omnibus database under accession code GSE270553. The flow cytometry fcs files generated in this study have been deposited in Flow Repository Data base under access code FR-FCM-Z8F2. The Luminex data and processed data generated in this study are provided in Source Data file. All other data are available in the article and its Supplementary files or from the corresponding author upon request. Source data are provided with this paper.

## Code availability

Code used to analyse RNAseq data is available https://github.com/Boyle-Lab-CRDV/Age-is-an-intrinsic-driver-of-inflammatory-responses-to-malaria/.

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

## Acknowledgements

We thank all participants in this study; the clinical and laboratory research staff and the hospital directors at the study sites, the head of Clinical Research Centre Malaysia and colleagues and staff of the UK medical research council *P. knowlesi* Monkeybar Project. RBC used for parasite culture were provided by Australian Red Cross Blood Bank (Brisbane). We thank the Director-General of Health, Malaysia, for per-mission to publish this article. This work was supported by the National Health and Medical Research Council of Australia (program grant 1132975 to C.R.E and N.M.A); Senior Research Fellowships C.R.E (1154265) and N.M.A (1135820), Career Development Award 1141278, Project Grant 1125656, and Ideas Grant 1181932 to M.J.B.; Program Grant 290208, Emerging Leadership 2 Fellowships to B.E.B and M.J.G); the CSL Centenary Fellowship to M.J.B, and the Snow Medical Foundation Fel-lowship 2022/SF167 to M.J.B. The Burnet Institute is supported by the NHMRC for Independent Research Institutes Infrastructure Support Scheme and the Victorian State Government Operational Infrastructure Support.

## Author contributions

Conceptualization and methodology: J.R.L., N.L.D., G.M., M.J.B. Investi-gation and validation: N.L.D., J.R.L., D.A., A.S., P.T., B.E.B., M.G., M.J.B. Formal analysis: N.L.D., J.R.L., Z.P., M.G., M.J.B. Resources: P.T., P.B., K.P., G.M., T.W., B.E.B., M.G., N.M.A. Data Curation: N.L.D., J.R.L., Z.P., M.G., M.J.B. Writing - Original Draft: N.L.D., J.R.L., M.G., M.J.B. Supervision: J.A.L., C.E., N.M.A., T.W., M.G., B.E.B., M.J.B. Writing - Review & Editing: N.L.D., J.R.L., G.M., M.J.B. with critical feedback and approval from all authors. All authors have read and approved the final version of the manuscript.

## Competing interests

The authors declare no competing interests.
