## [Transparent Peer Review file · Nature Communications]

Age is an intrinsic driver of inflammatory responses to malaria

Corresponding Author: Dr Michelle Boyle

Version 0:

Reviewer comments:

Reviewer #1

(Remarks to the Author)

In this manuscript Loughland et al address an important question in malaria research, which has so far been studied relatively little, namely the intrinsic effect of age (i.e. when uncoupled from prior exposure) on the immune response to malaria.

They address this using ex vivo samples from previously described cohorts of symptomatic paediatric and adult (uncomplicated and severe) malaria cases in Malaysia, where transmission levels are so low that many adults are expected to be malaria-naïve. In addition they assessed innate immune responses to blood-stage malaria parasites in vitro, in PBMCs from malaria-naïve Australian children and adults being followed at an allergy clinic, focussing in particular on the innate(-like) leukocyte populations monocytes and $\gamma\delta$ T cells.

They show that the adults in their cohorts with clinical malaria tend to have higher circulating levels of inflammatory chemokines and other markers, associated with more clinical symptoms. Monocytes and $\gamma\delta$ T cells of malaria-naïve adults tend to produce a stronger innate inflammatory response to blood-stage malaria parasites in vitro, compared to those of children. Interestingly, responses by adult monocytes also tended to be more monofunctional compared to those by paediatric monocytes.

Finding suitable samples to address this question is difficult due to the correlation between age and prior exposure in most (highly) endemic settings, which is likely the main reason why this question been relatively understudied. The Malaysian cohorts represent a good setting to address the question, and indeed have been used by the authors to do so previously, although some potential for bias remains (see below). The Australian cohort appears to be relatively free of bias, although it is not described what the underlying (allergological/immunological) diagnoses were in the children and adults, which may differ by age of presentation and affect immune responses to other triggers, e.g. malaria parasites.

The laboratory methodologies used, e.g. multiplex bead-array, PBMC stimulation and flow cytometry are considered routine in this line of research and appear sound, although I am insufficiently experienced in transcriptional/RNAseq analysis to assess this critically. The manuscript is well-written and concise and the figures generally clear and interpretable.

I have a few remaining points/questions regarding interpretation:

Any study involving passive selection of participants is potentially subject to bias. In the Malaysian cohort, were the adult and paediatric patients similar in terms of their clinical course, in particular the duration illness before presenting to hospital (and being included/sampled)? It is quite conceivable that the symptom threshold for concerned parents to bring their sick child to hospital is different (lower) than that of adults (particularly bread-winners) to present to hospital themselves. This could explain not only the higher symptom score, but also the stronger systemic inflammation in the adult patients compared to the children. Do they authors have any additional information to help assess this potential bias?

Just out of curiosity, how do the authors explain the high proportion of males in the Malaysian cohort: are males more likely to become infected (for men vs women this might be due to gender-related behaviour patterns that increase risk of (environmental) exposure, but for boys vs girls this would be less obvious), are they more likely to become ill if infected ("man-flu" or genuinely more susceptible), are they more likely to (be) present(ed) to hospital if they fall ill (which has been shown previously for male vs female children in resource-poor societies that distinguish male vs female economic value), or were they more likely to have sufficient remaining stored plasma volume for inclusion into this particular study?

It is not quite clear how multiplex analytes below the LLOQ were dealt with. Was the LLOQ-1 calculated on a linear scale (e.g. if LLOQ=1.5 pg/mL then LLOQ-1=0.5 pg/mL) or a log scale (e.g. if LLOQ=-0.3log10 pg/mL then LLOQ-1=-1.3log10 pg/mL)? And was this calculated before or after correcting for the (3x) dilution factor of the original samples? The large

number of data points plotted at exactly 1×10^{-3} pg/mL in Fig 1C suggest that these all fell below the LLOQ (which must have been 1.001 pg/mL or 1×10^{-2} pg/mL, depending on whether a linear or log subtraction was applied), but there is a notable absence of data points between 1×10^{-3} and 1×10^0 pg/mL (except for a single conspicuous CXCL9 value??), suggesting that the assay cannot actually generate values below 1 pg/mL. All this is not hugely problematic (and indeed a commonly encountered issue), but potentially more so are statistical analyses on data sets in which so many data points have been assigned arbitrary values. In particular, it appears questionable whether (even) a (non-parametric) correlation analysis between age and CCL3 or CCL8 is valid. And in general, the correlations between age and analytes appear to be driven by a disproportionate number samples with values <LLOQ in patients <20 years without any apparent further age correlations above that age (and even within the population of <20 year-olds, any age-correlations are not particularly obvious except perhaps for CRP). Were there any differences in hospital sample collection (tube type or size, etc.) or processing/storage practices in children/adolescents vs adults that might have subtly reduced analyte measurements? Was CMV status assessed for the Malaysian cohorts? And were immune responses in the malaria-naïve PBMC donors associated with their CMV status?

Were any other (non-specific or specific stimuli) used with the malaria-naïve PBMCs? It would be interesting to see if the age-related differences in response to stimulation are specific to blood-stage malaria parasites (presumably not), or also occur in response to other classes of pathogens that are recognised through innate pattern recognition receptors.

The apparent age-related decrease in monocyte polyfunctionality is intriguing. Has this ever previously been seen? Could the authors speculate on potential underlying mechanisms?

Could the authors speculate on the implication of their findings on age vs pathogenicity in the context of the WHO's recent recommendation to provide the pre-erythrocytic malaria vaccines RTS,S and R21 to infants in (highly) endemic countries? The duration of protective efficacy of both vaccines is limited and the jury is still out on how many yearly boosters children are to receive – one or two, or potentially up to 5 years of age (but presumably not life-long). In any case, vaccinated individuals will tend to temporarily experience fewer blood-stage infections, delaying the acquisition of (tolerising?) blood-stage immunity. Could this lead not only to an age-shift in the burden of clinical malaria, but also (based on the authors' findings) to an overall increase in the clinical severity of malaria?

Reviewer #2

(Remarks to the Author)

In this study Loughland and Dooley et al., have investigated the effect of age on the inflammatory response (focusing especially on monocytes and gdT cells) following stimulation with uRBC and pRBC. They could show that cells from adults responded more strongly with inflammatory mediators/signal pathways compared to cells from children. These data are important as they progress our understanding of why malaria can have different impact on people of different ages.

Overall, the manuscript is well written with nicely presented data/visualizations.

General comments

I enjoyed the experimental approaches in malaria-naïve individuals to assess the role of age separate from potential malaria exposure.

In the analysis of monocytes (Figure 3) the DEG analysis points to a much stronger activation of the adult monocytes. But you also mention that the children have a higher baseline activation (as you also indicate by connecting with the cytokine data). Could the high baseline expression of potentially inflammatory responses in children impact the comparison when primarily investigating DEGs? Is it so that adults respond much more vigorously, or are they just reaching a similar level? Could read counts be compared?

Regarding the presentation/selection of data from Figure 3E-H, I understand the selection, but it is also a selection to fits a narrative quite strongly. There are also almost equally high IPA data associated with Th2 and IL4/5 signalling in adults. I also understand that there were few pathways for children, but when looking at the receptors, cytokines etc, it seems to me to also represent a quite strong inflammatory profile associated with type I IFNs, TLR9 and 3 signalling, which I found interesting. Any thoughts on that?

I was a bit surprised by the contrasting pattern of gdT cells in adults/children compared with monocytes in adults/children. Again here though, how much do you think the impact of the baseline affects the readouts, which are based on DEGs?

The PCA indicates some variation between donors. Especially at baseline. All DEG data is generated based no pooled data, I guess, but what does the data look like at an individual donor level (since you have run 5 of each donor/stim, it should be possible to look at this)? It could be interesting to also plot with count data (or some kind of ratio stim/baseline) for each donor.

I think the discussion on TLRs was interesting in the context of age. I wondered though why TLR9 was not mentioned since its also been associated with responses to malaria and also heavily associated with for example IL10 production from B cells (which I agree is not necessarily relevant here though). <https://www.pnas.org/doi/10.1073/pnas.0608745104>

Speaking of TLR stim, I was wondering if sex might have an impact. I didn't see sex discussed anything, but I imagine it could play a role for example in TLR7/8 signalling. Were the groups used for the transcriptomic analysis sex-matched (I saw they were in the tables) for the overall larger groups.

In the discussion (lines 410-413) you mention a connection with Tfh cells. I notice that you have all the markers needed to investigate them in the panel for Figure 6. What was the rationale for including Treg data but not Tfh data?

One thing that was not really discussed was the potential impact of the overall cell composition in the stimulated PBMCs from the young and adult donors. You thoroughly investigated monocytes and gdT cells, but they were stimulated in the context of other cells (which differ quite a bit between young and adults). Could this have made a difference?

Specific comments

Line 103: Please also indicate age range.

Line 107: indicate according to which criteria they were defined as severe disease (i see noted in table, but since not in main, it's not very accessible). What is the reason for 2014 criteria and not more recent?

Line 132: difference

Figure 1: purely from an aesthetic point of view, why are the dots in Fig1A not round, but ovals (considering later round dots)? Makes the figure look manually stretched, but also not.

For the monocyte gate after stimulation (lines 167-170), could mDCs also end up with the non-classical or are they CD86 neg? I guess the intermediate monos will end up with the classical (if it's just about losing CD16)?

Line 182: Figure 2D

Suppl Fig 8: It would be nice to add a histogram or such at the end of (A) to indicate FoxP3 levels in CD25+CD127lo vs CD25- CD4 T cells.

Line 406: beneficial?

Reviewer #3

(Remarks to the Author)

The manuscript documents a well-conducted study that is methodologically sound, well-written and very well-presented. The following comments identify several areas that the authors should consider addressing in order to enhance its overall quality.

Methods/Study design:

1. It appears that the authors created a custom multiplex for assessment of the described panel of analytes in cryopreserved plasma samples from the Malaysian patients. Given (i) their known role in the inflammatory response to *P. falciparum* infection, and (ii) the fact that they were included in the flow cytometric and transcriptional profile assessments conducted as part of the in vitro experiments, it is surprising that neither IL-1b, IL-6, IL-12 nor TNF were included in the custom panel of analytes assessed in plasma here. The rationale for this choice needs to be clarified by the authors for readers who will wonder why there is not the overlap across the different elements of the study that would seem to make logical sense in this context.

2. Blood donors for PBMC collection were recruited from amongst individuals presenting at an outpatient allergen clinic. Although deemed to be healthy - by an on-site immunologist, as indicated - one could legitimately ask why these particular donors were actually attending the clinic, if indeed they had no clinical symptoms. The authors should clarify this point to remove any doubt concerning the donors' clinical status/the status of their PBMC prior to use in in vitro experiments.

3. Specific conditions are described for the in vitro stimulation of innate cells with pRBC i.e. a 1:1 ratio, with co-culture for 24 hours, as well as, separately, for Treg/CD4+ T cells (3:1 ratio, co-culture for 5 days). The authors need to clarify how and why these particular conditions were chosen - presumably through a process of optimisation for the proposed assessments? Have the authors used the same conditions previously in other (published or unpublished) studies? Please clarify.

4. The in vitro part of the authors' study focused on two types of innate immune cells in their assessments of cellular responses to pRBC, namely (classical, CD14+) monocytes and gd T cells. A third type of innate immune cells present in PBMC, namely NK cells, have long been thought to play a pivotal role in the human innate immune response to *P. falciparum*, begging the question of why the authors chose to neglect them in their study. Here again, I feel, they should justify their choice for greater clarity.

Results & Discussion

1. With respect to the data generated on monocytes subjected to pRBC stimulation in vitro:

(i) intracellular staining, as depicted in Figure 2B, revealed a significantly higher % of adults' monocytes with CCL2, and a significantly higher % of childrens' monocytes with IL10. Otherwise, with intracellular staining, there were no differences concerning the % monocytes with any of the other pro-inflammatory cytokines assessed by this method (IL1b, IL6 or TNF). At the transcriptional level, however, the picture appears quite different. Here, as depicted in Figure 3G, both IL1b and TNF genes are highly upregulated in adults' monocytes compared to childrens', along with several other cytokine genes that showed a similar profile, but apparently not CCL2, results that clearly do not align with those reported from intracellular staining. The transcriptional profile of IL10 did reveal upregulation to a greater extent in childrens' monocytes, so consistent here with the results obtained through intracellular staining. Somewhat counter-intuitively, however, given its known association with IL10 production, Figure 3E depicts the macrophage alternative activation signaling pathway as being upregulated in adults' monocytes but - in counterpoint to the observations mentioned above - downregulated in childrens'. The latter observation appears itself to contrast directly with the finding (Lines 242-243), in childrens' monocytes, of the upregulation of transcriptional factors, that included HOXA3, which inhibit M1 polarization and drive M2 (alternatively activated) polarization in monocytes. The interpretation of these apparently contrasting data, including the relative weight given to the results of intracellular staining versus those from transcriptional profiling, seems to me not so clear cut as the authors' statement (Lines 255-257) indicates: 'transcriptional and cytokine production data show that classical monocytes' responsiveness to malaria is age dependent, with enhanced inflammatory responses in adults compared to children in malaria naive individuals.' The disparities in the findings from the different data sets, as are pointed out here, suggest to me that a rather more nuanced interpretation is warranted than the one the authors have presented.

(ii) There are other aspects of the data, in particular those concerning childrens' monocyte activity, that merit some attention, currently missing, in the discussion, in my opinion. It is striking, for example, that the basal (uRBC condition) % of childrens' cytokine containing cells, with the exception of CCL2, was markedly higher than that of adults (Figure 2A). In addition, as the authors point out, it was notable that childrens' monocytes had a greater propensity for polyfunctionality in terms of their content of more than 1 cytokine following pRBC stimulation (Figure 2C,D). Could the ex vivo analyses the authors performed shed any light on this particular aspect e.g. differences in the ex vivo activation status of childrens' versus adults' cells? Childrens' classical monocytes did, for example, display significantly higher levels of CD86 than did adults' monocytes (Supplementary Figure 2G), with the same being true for their different DC populations. Notable also was the significantly higher level of expression of ICOS on childrens' CD4+ T cells (Supplementary Figure 2I). These indications of apparently higher levels of (pre-existing) activation of different cell types in children warrant some discussion.

(iii) Following on from the above, from a purely practical/technical perspective, the culture conditions employed would, I conjecture, have allowed for schizont rupture at a certain point, and hence exposure of PBMC to the full array of parasite-derived products, obviously including hemozoin, DNA, extracellular vesicles, as well as to intact free merozoites, all of which could be expected to influence the outcomes measured via the involvement of TLR-mediated interactions, as well as via other receptor/signaling pathways. The authors should offer some comment on and discussion of these aspects, whilst also considering the possibility that trained immunity might be playing a role here. Such a phenomenon could potentially explain the apparently age-related upregulation of the expression of pRBC-mediated TLR4 (higher in adults' monocytes) identified in this study i.e. the possibility that adults' responses to pRBC in vitro might be skewed due to prior/life-long exposure to other pathogens.

(iv) Figure 3 is entitled 'Increased transcriptional activation in response to *P. falciparum* in monocytes from malaria naive adults compared to children', whilst Supplementary Figure 5 is entitled 'IPA analysis of monocyte genes which were higher in children compared to adults after primary pRBC stimulation'. For the uninitiated reader, and to avoid confusion in interpretation, I suggest that the authors modify the latter title to read, for example, 'Pathway and upstream regulator analyses of monocyte genes which were higher in children compared to adults after primary pRBC stimulation'.

(iv) Figure 3H appears to lack the data relating to expression of one of the genes (Fas?) in childrens' cells

2. In Lines 364-373 the authors discuss the potential role for different TLR in the pRBC-induced monocyte responses measured. Whilst they correctly mention TLR3, TLR4, TLR7 & TLR8, they fail to mention TLR9 in the same context. There is evidence in the published literature identifying specific interactions between TLR9 and complexes of hemozoin/parasite-derived DNA. This omission should be corrected for completeness.

3. In comparison to the data pertaining to monocyte activity, as discussed above, those pertaining to gdT cell activity are less contentious in the sense that childrens' and adults' cells responded to pRBC stimulation in broadly similar ways, albeit with significantly higher % of adult cells containing IFN γ and/or TNF. There is a wealth of evidence in the literature supporting a major role for gd T cells in the innate immune response to *P. falciparum* that these data serve to underscore. The data concerning CD4+ T cells (Treg and others) were unremarkable in the sense that, under the chosen in vitro conditions, childrens' and adults' profiles were very similar. In the context of the latter, the authors might comment on the biological relevance of a pRBC:PBMC ratio of 3:1.

4. Statements such as 'monocytes and V δ 2+ γ δ T cells produced significantly higher inflammatory cytokines in adults compared to children following in vitro parasite stimulation' (Discussion lines 349-351, and several similar statements in Results) are misleading in the sense that they imply - at least to me - the quantification of cytokines in culture supernatants, whereas they are actually referring to the % cells detected as positive by intracellular staining and/or transcriptional profiles. For clarity, these statements should be reworded to better reflect the true nature of the data collected.

Conclusions

1. Whilst the in vitro studies performed here are clearly informative, although nevertheless tempered by the specific co-culture conditions chosen, I think it is important for the authors to point out - possibly in their conclusion and/or in the paragraph on their study limitations - that primo-exposures in vivo, whether in children or in adults, will necessarily involve initiation of immunological responses by pivotal early-responding cell types, such as dendritic cells, that were not the focus of the study here. Evidence supportive of a role for DC in mediating responses to asexual blood stages can be found, for example, in the published literature emanating from in vivo (CHMI) studies of primo-infections with *P. falciparum* in naive adults (e.g. Teirlinck et al, *Infection & Immunity*, 2015).

Reviewer: Adrian J F Luty

Version 1:

Reviewer comments:

Reviewer #1

(Remarks to the Author)

I thank the authors for their considerate responses to my queries, which I feel are now sufficiently addressed.

I have one very small new point, which the authors could consider revising (or not) in a final version, but would not require further review: the IPTi trial by Muhindo et al. quoted by the authors in the Discussion compared two different frequencies of DHA-PPQ IPTi (every 4 vs every 12 weeks), rather than IPTi vs no IPTi. The incidence of malaria episodes in the follow-up year after cessation of IPTi was lower in children who had received 4-weekly DHA-PPQ compared to those who had received it only 12-weekly. This still suggests that reducing the incidence of infection in younger children (e.g. by vaccination) has beneficial rather detrimental consequences in older children, but strictly speaking it cannot be surmised (as the authors seem to suggest) that children who had not received any IPTi would have had a(n even) higher incidence of malaria during this follow-up year than the (12-weekly) IPTi recipients.

(Remarks on code availability)

Reviewer #2

(Remarks to the Author)

I think the authors have sufficiently addressed all my comments and I have no further queries regarding the manuscript.

(Remarks on code availability)

The code is well annotated and easy to read with no obvious errors. The readme contains information about packages and versions. However, I did not install and run the code to look for errors.

Reviewer #3

(Remarks to the Author)

The revised version of the manuscript has taken into account my comments largely to my satisfaction.

I have just one outstanding issue concerning methodology that the authors should address:

As they will surely know, cryopreservation methods exist that would have allowed for the authors to have used pRBC containing live, viable parasites for in vitro stimulations as opposed to the non-viable pRBC resulting from the glycerolyte-based method chosen. The use of viable parasites would, I suggest, have more authentically replicated in vivo conditions. Do they know, or can they at least conjecture, what the condition was (ruptured/lysed/intact throughout?) of the non-viable pRBC they used over the period of culture in either 1- or 5-day stimulations?

The reasons for the choice they made in this context should be explained for clarity, and the fact that they used non-viable pRBC for the stimulations should also be included as a further limitation of their study.

Reviewer Adrian J F Luty

(Remarks on code availability)

We thank the reviewers for their time and insights. We have addressed all
comments as outlined below to strengthen our manuscript. Page numbers
indicated in the rebuttal are for track changes version of resubmission.

**Reviewer #1 (Remarks to the Author)**

*In this manuscript Loughland et al address an important question in malaria research,*
*which has so far been studied relatively little, namely the intrinsic effect of age (i.e.*
*when uncoupled from prior exposure) on the immune response to malaria.*

*They address this using ex vivo samples from previously described cohorts of*
*symptomatic paediatric and adult (uncomplicated and severe) malaria cases in*
*Malaysia, where transmission levels are so low that many adults are expected to be*
*malaria-naïve. In addition they assessed innate immune responses to blood-stage*
*malaria parasites in vitro, in PBMCs from malaria-naïve Australian children and adults*
*being followed at an allergy clinic, focussing in particular on the innate(-like) leukocyte*
*populations monocytes and $\gamma\delta T$ cells.*

*They show that the adults in their cohorts with clinical malaria tend to have higher*
*circulating levels of inflammatory chemokines and other markers, associated with more*
*clinical symptoms. Monocytes and $\gamma\delta T$ cells of malaria-naïve adults tend to produce a*
*stronger innate inflammatory response to blood-stage malaria parasites in vitro,*
*compared to those of children. Interestingly, responses by adult monocytes also tended*
*to be more monofunctional compared to those by paediatric monocytes.*

*Finding suitable samples to address this question is difficult due to the correlation*
*between age and prior exposure in most (highly) endemic settings, which is likely the*
*main reason why this question been relatively understudied. The Malaysian cohorts*
*represent a good setting to address the question, and indeed have been used by the*
*authors to do so previously, although some potential for bias remains (see below). The*
*Australian cohort appears to be relatively free of bias, although it is not described what*
*the underlying (allergological/immunological) diagnoses were in the children and adults,*
*which may differ by age of presentation and affect immune responses to other triggers,*
*e.g. malaria parasites.*

*The laboratory methodologies used, e.g. multiplex bead-array, PBMC stimulation and*
*flow cytometry are considered routine in this line of research and appear sound,*
*although I am insufficiently experienced in transcriptional/RNAseq analysis to assess*
*this critically. The manuscript is well-written and concise and the figures generally clear*
*and interpretable.*

We thank the reviewer for their time and insights. We have addressed all points
as outlined below. Additionally, the underlying allergy diagnosis in the malaria
naïve Australian cohort is now included in Supplementary Table S3 and methods
section lines 590-592.

*I have a few remaining points/questions regarding interpretation:*

*Any study involving passive selection of participants is potentially subject to bias. In the*
*Malaysian cohort, were the adult and paediatric patients similar in terms of their clinical*

*course, in particular the duration illness before presenting to hospital (and being*
*included/sampled)? It is quite conceivable that the symptom threshold for concerned*
*parents to bring their sick child to hospital is different (lower) than that of adults*
*(particularly bread-winners) to present to hospital themselves. This could explain not*
*only the higher symptom score, but also the stronger systemic inflammation in the adult*
*patients compared to the children. Do they authors have any additional information to*
*help assess this potential bias?*

We agree with the reviewer that clinical course and presentation at hospital may
differ between children and adults. While we cannot formally rule this difference
out, the number of days of fever prior to presentation was recorded and did not
associate with age, nor differ between children and adults. This information is
now included in Supplementary Table S2, and in lines 117-119, and 142-143 of
results.

*Just out of curiosity, how do the authors explain the high proportion of males in the*
*Malaysian cohort: are males more likely to become infected (for men vs women this*
*might be due to gender-related behaviour patterns that increase risk of (environmental)*
*exposure, but for boys vs girls this would be less obvious), are they more likely to*
*become ill if infected (“man-’flu” or genuinely more susceptible), are they more likely to*
*(be) present(ed) to hospital if they fall ill (which has been shown previously for male vs*
*female children in resource-poor societies that distinguish male vs female economic*
*value), or were they more likely to have sufficient remaining stored plasma volume for*
*inclusion into this particular study?*

For all Malaysian cohorts, a higher proportion of infected males compared to
females is observed for all malaria species, possibly due to increased exposure
to forest settings. This information is now included in methods lines 572-574 and
noted as a limitation in the discussion line 517-522. While we have hypothesised
that this is due to forest workers (occupationally associated with males), it is also
possible that other environmental factors contribute, particularly in children.

*It is not quite clear how multiplex analytes below the LLOQ were dealt with. Was the*
*LLOQ-1 calculated on a linear scale (e.g. if LLOQ=1.5 pg/mL then LLOQ-1=0.5 pg/mL)*
*or a log scale (e.g. if LLOQ=0.3log₁₀ pg/mL then LLOQ-1=-1.3log₁₀ pg/mL)? And was*
*this calculated before or after correcting for the (3x) dilution factor of the original*
*samples? The large number of data points plotted at exactly 1x10⁻³ pg/mL in Fig 1C*
*suggest that these all fell below the LLOQ (which must have been 1.001 pg/mL or*
*1x10⁻² pg/mL, depending on whether a linear or log subtraction was applied), but there*
*is a notable absence of data points between 1x10⁻³ and 1x10⁰ pg/mL (except for a*
*single conspicuous CXCL9 value??), suggesting that the assay cannot actually*
*generate values below 1 pg/mL. All this is not hugely problematic (and indeed a*
*commonly encountered issue), but potentially more so are statistical analyses on data*
*sets in which so many data points have been assigned arbitrary values. In particular, it*

*appears questionable whether (even) a (non-parametric) correlation analysis between*
*age and CCL3 or CCL8 is valid. And in general, the correlations between age and*
*analytes appear to be driven by a disproportionate number samples with values <LLOQ*
*in patients <20 years without any apparent further age correlations above that age (and*
*even within the population of <20 year-olds, any age-correlations are not particularly*
*obvious except perhaps for CRP). Were there any differences in hospital sample*
*collection (tube type or size, etc.) or processing/storage practices in*
*children/adolescents vs adults that might have subtly reduced analyte measurements?*

To clarify, the LLOQ/ULOQ is specific to each analyte, based on analyte specific
standards and regression fits. For analyte concentrations below the LLOQ we
have imputed them as analyte LLOQ – 1, with a minimum of 0.001 value, and
data above ULOQ we imputed as ULOQ+1.

In the original submission, we had calculated a LLOQ prior to batch correction,
resulting in slightly different imputed values across batches the outlier of CXCL9
noted by reviewer. For clarity in revision, we have slightly modified data analysis
to use a matched LLOQ/ULOQ across all data, and re-generated Figure 1 and
Supplementary Figure 1. In addition, we have highlighted data that is imputed as
LLOQ or ULOQ within the data. Modified analysis did no impact findings, but we
hope increases clarity. We have also modified methods lines 623-625.

Further, as noted by reviewer, there is for some analytes a disproportionate
number of samples with values <LLOQ which have been imputed as zeros/0.001
in children. There were no differences in hospital sample collections or
processing between adults and children, who were all enrolled at the same study
sites. As such, we interpret this finding to be due to children's overall lower
responses. While we have analysed these relationships with Spearman's
correlations, a non-parametric ranking approach that should be less impacted by
zero inflated data, we agree with the reviewer that the correlations reported may
be driven by disproportionate <LLOQ values in children. We have highlighted this
limitation in the dynamic range of Luminex at lower concentrations in discussion
lines 522-529.

*Was CMV status assessed for the Malaysian cohorts? And were immune responses in*
*the malaria-naïve PBMC donors associated with their CMV status?*

CMV sero-status was not tested in this study. However, we have previously
reported 100% CMV sero-positivity in a cohort of 43 individuals the same study
area (Dooley et al, Nat Coms, 2023), thus assume that CMV status will be very
high. This information is now included in lines 599-602 of methods.

In the malaria-naïve cohort, we did not see any significant differences between
CMV status and immune responses. For the reviewer's interest, this analysis is
below. We have not included this data in the re-submission but are happy to

include as a supplementary if requested. CMV status is included in the excel files
for all data uploaded with submission thus is available for individual researcher
analysis.

A

B

**Figure:** Association between cytokine production following in vitro parasite
stimulation and cytomegalovirus serostatus.
Proportion of Pf activated cytokine expressing classical monocytes (cytokine
positive frequency in pRBC condition subtracted by uRBC condition), in
monocytes (A) or V δ 2⁺g δ T cells (B) in CMV negative and positive individuals. P
is Wilcoxon signed rank test.

*Were any other (non-specific or specific stimuli) used with the malaria-naïve PBMCs? It*
*would be interesting to see if the age-related differences in response to stimulation are*
*specific to blood-stage malaria parasites (presumably not), or also occur in response to*
*other classes of pathogens that are recognised through innate pattern recognition*
*receptors.*

Due to limited sample and cell number availability, we were unable to test
responses to other stimuli. We agree with the reviewer that age-related
differences are not specific only to malaria parasites and are likely to be seen
against other specific or non-specific stimuli. We have discussed that the age
dependent response to malaria may be driven by specific pattern recognition
pathways (lines 414-419). There is evidence that the malaria parasite can
engage TLR2/4 (GPI anchors), TLR8 (Plasmodium RNA) and TLR9 (hemozoin
and/or malaria derived DNA). Responses to TLR2, TLR4 and TLR9 are reported
to be age dependent, while TLR7/8 are not.

*The apparent age-related decrease in monocyte polyfunctionality is intriguing. Has this*

*ever previously been seen? Could the authors speculate on potential underlying*
*mechanisms?*

To the best of our knowledge, there are no previous reports of age dependent
changes to polyfunctional monocytes between children and adults comparable to
our study. However, increased polyfunctional inflammatory monocytes have been
associated with aging (>60 years) and CMV infection
(doi:[10.1093/gerona/glv121](https://doi.org/10.1093/gerona/glv121)), and are expanded in older HIV infected individuals
(doi:[10.1093/infdis/jiv520](https://doi.org/10.1093/infdis/jiv520)).

Regarding potential mechanisms underpinning age dependent changes to
malaria responsiveness, one possibility is trained immunity in adult monocytes
and gd T cells which have been exposed to a greater accumulation of infections.
This is now included in discussion lines 460-484.

*Could the authors speculate on the implication of their findings on age vs pathogenicity*
*in the context of the WHO's recent recommendation to provide the pre-erythrocytic*
*malaria vaccines RTS,S and R21 to infants in (highly) endemic countries? The duration*
*of protective efficacy of both vaccines is limited and the jury is still out on how many*
*yearly boosters children are to receive – one or two, or potentially up to 5 years of age*
*(but presumably not life-long). In any case, vaccinated individuals will tend to*
*temporarily experience fewer blood-stage infections, delaying the acquisition of*
*(tolerising?) blood-stage immunity. Could this lead not only to an age-shift in the burden*
*of clinical malaria, but also (based on the authors' findings) to an overall increase in the*
*clinical severity of malaria?*

We agree that data could suggest that shifting ages of exposure to malaria may
impact disease severity. However, other data suggest a more nuanced impact,
and other factors including environment and pathogen exposure may also be
important. These points are now included in discussion lines 485-488 and 494-
500.

**Reviewer #2 (Remarks to the Author)**

*In this study Loughland and Dooley et al., have investigated the effect of age on the*
*inflammatory response (focusing especially on monocytes and gdT cells) following*
*stimulation with uRBC and pRBC. They could show that cells from adults responded*
*more strongly with inflammatory mediators/signal pathways compared to cells from*
*children. These data are important as they progress our understanding of why malaria*
*can have different impact on people of different ages.*

*Overall, the manuscript is well written with nicely presented data/visualizations.*

**General comments**

*I enjoyed the experimental approaches in malaria-naïve individuals to assess the role of*
*age separate from potential malaria exposure.*

We thank the reviewer for their comments and appreciation of our data.

*In the analysis of monocytes (Figure 3) the DEG analysis points to a much stronger*
*activation of the adult monocytes. But you also mention that the children have a higher*
*baseline activation (as you also indicate by connecting with the cytokine data). Could*
*the high baseline expression of potentially inflammatory responses in children impact*
*the comparison when primarily investigating DEGs? Is it so that adults respond much*
*more vigorously, or are they just reaching a similar level? Could read counts be*
*compared?*

A negative binomial distribution via regression models of normalized count data
and Wald test was used to compare gene expression in paired pre- and post-
stimulated samples from children and adults. The model analyses the impact of
age (child versus adult), stimulation (pre-stim versus post-stim) and the
interaction between age and stimulation, simultaneously. The model also
accounts for a random effect of each individual and the paired nature of the data.
In revision, we have edited methods lines 703-706 for clarity.

With this analytical approach, identified DEGs, take into account the baseline
responses, and thus should not be biased by the higher inflammatory levels at
baseline in children. Instead, this model shows that adults respond more
vigorously. In revision, we have highlighted these data by more clearly defining
the genes of interest in results line 225-232, and including an additional
Supplementary Figure 6A-B, to visualise the modelled gene expression at
baseline and post-stimulation in genes of interest in Figure 3D. This is a more
appropriate visualisation of the data rather than counts, due to the regression
normalisation of count data prior to analysis in the model.

*Regarding the presentation/selection of data from Figure 3E-H, I understand the*
*selection, but it is also a selection to fits a narrative quite strongly. There are also almost*

*equally high IPA data associated with Th2 and IL4/5 signalling in adults. I also*
*understand that there were few pathways for children, but when looking at the*
*receptors, cytokines etc, it seems to me to also represent a quite strong inflammatory*
*profile associated with type I IFNs, TLR9 and 3 signalling, which I found interesting. Any*
*thoughts on that?*

We have made edits to the results section to highlight upregulation of the Th2
associated pathways in adults (lines 269-272) and inflammatory pathways in
children (lines 260-262, and 275-277). Due to the relatively low z-score identified
in transmembrane receptor analysis for TLR9 and TLR3 we have chosen not to
comment on this finding as we feel it may be due to DEGs overlapping across
multiple pathways. In addition, we have highlighted that pathway analysis
findings required specific validation in future studies in lines 283-285.

*I was a bit surprised by the contrasting pattern of gdT cells in adults/children compared*
*with monocytes in adults/children. Again here though, how much do you think the*
*impact of the baseline affects the readouts, which are based on DEGs?*

As described above, the glmmSeq analysis considers baseline differences when
identifying DEGs in response to stimulation. Thus, while baseline age dependent
differences are identified in $\gamma\delta$ T cells, these should not impact the DEGs
identified within the model.

Regarding the contrasting patterns of $\gamma\delta$ T cells and monocytes in RNAseq
analysis, we were also surprised by this result, particularly as we detected clear
differences in IFN γ and TNF production in $\gamma\delta$ T cells at the protein level. While we
are unable to explain these differences, they may be due to different timing of
stimulation between assays or point to post-translational regulation of cytokine
production as indicated in lines 355-357. In revision we have also discussed
these differences in discussion lines 434-438.

*The PCA indicates some variation between donors. Especially at baseline. All DEG data*
*is generated based no pooled data, I guess, but what does the data look like at an*
*individual donor level (since you have run 5 of each donor/stim, it should be possible to*
*look at this)? It could be interesting to also plot with count data (or some kind of ratio*
*stim/baseline) for each donor.*

Biological variation among donors is expected in bulk RNA-seq studies. In our
data, we did not observe abnormal overdispersion, which allowed us to model
gene-level expression robustly and identify (DEGs). Importantly, our statistical
approach accounts for donor-specific variation: we used glmmSeq, a generalized
linear mixed model framework, which incorporates individual donor as a random
effect. This enables the partitioning of within-donor and between-donor variability,
reducing residual noise and improving the reliability of DEG identification.

As indicated above, while downstream analysis of DEGs is on the pooled data,
DEGs are identified with glmmSeq analysis which takes into account each
donors paired data. This generalised model allowed the identification of DEGs
that were significant with age and/or stimulation, taking into account underlying
donor heterogeneity.

*I think the discussion on TLRs was interesting in the context of age. I wondered though*
*why TLR9 was not mentioned since its also been associated with responses to malaria*
*and also heavily associated with for example IL10 production from B cells (which I*
*agree is not necessarily relevant here*
*though). <https://www.pnas.org/doi/10.1073/pnas.0608745104>*

We agree with the reviewer that TLR9 is of potential importance, particularly in
the context of stimulation of total immune cells rather than isolated populations
and have now included in discussion lines 416-419.

*Speaking of TLR stim, I was wondering if sex might have an impact. I didn't see sex*
*discussed anything, but I imagine it could play a role for example in TLR7/8 signalling.*
*Were the groups used for the transcriptomic analysis sex-matched (I saw they were in*
*the tables) for the overall larger groups.*

We agree that sex is likely an important factor in both innate and adaptive
immune responses to malaria. In our study, we are not powered to investigate
sex related differences, and post-hoc analysis is discouraged by the journal when
study design is insufficient to enable meaningful conclusions. We have included
this limitation in lines 548-553. However, children and adults used in
transcriptomic analysis were sex-matched and this information is now included in
methods section line 682-683. In addition, sex is now included in Source Data
files so exploratory analysis can be performed by interested researchers.

*In the discussion (lines 410-413) you mention a connection with Tfh cells. I notice that*
*you have all the markers needed to investigate them in the panel for Figure 6. What was*
*the rationale for including Treg data but not Tfh data?*

We originally focused on Tregs as these cells have been shown to expand in
response to malaria stimulation in vitro and could regulate the increased
inflammatory responses observed in innate cells. In revision we have now
included Tfh responses along with effector CD4 T cell subsets for completeness.
These data are described in lines 381-391 and in Supplementary Figure 12. In
vitro activation of Tfh cells by parasite simulation were similar between adults and
children.

One thing that was not really discussed was the potential impact of the overall cell composition in the stimulated PBMCs from the young and adult donors. You thoroughly investigated monocytes and gdT cells, but they were stimulated in the context of other cells (which differ quite a bit between young and adults). Could this have made a difference?

We agree with the reviewer that cell composition differences (as seen in Supplementary Figure 2) could influence the monocyte and $\gamma\delta$ T cell responses via cell extrinsic ways. In revision, we have included new analysis of our data to explore correlations between baseline frequencies and activation signatures of cells with cytokine responses in monocytes and $\gamma\delta$ T cells. Data is included in Supplementary Figure 9. This analysis identified age- and cell type-specific associations between baseline immune cell frequencies or activation states and cytokine responses to *Pf*, however most associations were not significant. Notable associations included IL10 production by classical monocytes with baseline activation of cDCs and pDCs in children and TNF production by $V\delta 2^+$ $\gamma\delta$ T cells with baseline CD16 responses in B cells, CD4 T cells, cDCs, pDCs and non-classical monocytes. These data are now reported in results lines 319-330, and discussion lines 478-483. In addition, the importance of identifying cell intrinsic and extrinsic factors underpinning our findings is noted in lines 536-540.

Specific comments

Line 103: Please also indicate age range.

We have updated the text to include the age range (2-72). This is also included in Supplementary Table S1.

Line 107: indicate according to which criteria they were defined as severe disease (i see noted in table, but since not in main, it's not very accessible). What is the reason for 2014 criteria and not more recent?

To improve clarity, we have now included how severe disease was defined in the main manuscript, lines 110-111. The 2014 criteria is used as this was the approach in the parent cohorts leveraged for this study.

Line 132: difference

This error has been corrected line 136.

Figure 1: purely from an aesthetic point of view, why are the dots in Fig1A not round, but
ovals (considering later round dots)? Makes the figure look manually stretched, but also
not.

We have re-exported Figure 1A and have fixed this aesthetic error.

*For the monocyte gate after stimulation (lines 167-170), could mDCs also end up with*
*the non-classical or are they CD86 neg? I guess the intermediate monos will end up*
*with the classical (if it's just about losing CD16)?*

Myeloid DCs would be CD86 dim or negative compared to the monocytes, so
should be excluded from analysis. The reviewer is correct that intermediate
monocytes may also be in this gate, and this possibility is now indicated in lines
172-175.

*Line 182: Figure 2D*

Have updated the in-text figure reference to "Figure 2D".

*Suppl Fig 8: It would be nice to add a histogram or such at the end of (A) to indicate*
*FoxP3 levels*

As requested, we have updated Supplementary Fig 11A to include a FoxP3
histogram comparing FoxP3 levels between Tregs and total CD4 T cells.

*Line 406: beneficial?*

This error has been corrected – now line 490.

**Reviewer #3 Adrian J F Luty (Remarks to the Author):**

*The manuscript documents a well-conducted study that is methodologically sound, well-*
*written and very well-presented. The following comments identify several areas that the*
*authors should consider addressing in order to enhance its overall quality.*

We thank Prof Luty for his time and helpful feedback on our manuscript. We
have incorporated all suggestions as outlined.

*Methods/Study design:*

*1. It appears that the authors created a custom multiplex for assessment of the*
*described panel of analytes in cryopreserved plasma samples from the Malaysian*
*patients. Given (i) their known role in the inflammatory response to P. falciparum*
*infection, and (ii) the fact that they were included in the flow cytometric and*
*transcriptional profile assessments conducted as part of the in vitro experiments, it is*
*surprising that neither IL-1b, IL-6, IL-12 nor TNF were included in the custom panel of*
*analytes assessed in plasma here. The rationale for this choice needs to be clarified by*
*the authors for readers who will wonder why there is not the overlap across the different*
*elements of the study that would seem to make logical sense in this context.*

The custom multiplex kit used here was designed to include IL1b, IL6, IL12 and
TNF, along with other analytes (ANG1, IFN γ , IFN, IL4, IL18, RANKL, TRAIL,
TREM1, Granzyme-B, IL21, and OPG). Unfortunately, likely due to the multiplex
assay set up, some of these analytes, including IL1b, IL6, IL12 and TNF, had a
low number (<20%) of samples that fell below limit of quantification. It appears
that while we chose 3X dilution of plasma based on published protocols, that this
dilution was too high for many analytes. In the original submission, we removed
these analytes, along with Granzyme-B, IL18, IL21, and OPG, to focus on
analytes most commonly implicated in innate cell immunopathogenesis and
which had sufficient data in quantification limits.

In response to this query, and for further transparency, we have indicated
removed analytes that had a quantification rate lower than 20% across all
samples in methods lines 620-623. We have also included IL18, Granzyme-B,
IL21 and OPG in data analysis included in Figure 1.

*2. Blood donors for PBMC collection were recruited from amongst individuals presenting*
*at an outpatient allergen clinic. Although deemed to be healthy - by an on-site*
*immunologist, as indicated - one could legitimately ask why these particular donors*
*were actually attending the clinic, if indeed they had no clinical symptoms. The authors*
*should clarify this point to remove any doubt concerning the donors' clinical status/the*
*status of their PBMC prior to use in in vitro experiments.*

We agree that the immunological status of our malaria naïve cohorts is an
important consideration. To clarify, all participants were attending allergy clinic for
routine allergy and management. In revision, we have extended the information
of inclusion and exclusion criteria for this cohort in methods lines 585-594. In
addition, we have noted in limitations discussion that we cannot formally exclude
whether allergic status of donors may impact findings.

*3. Specific conditions are described for the in vitro stimulation of innate cells with pRBC*
*i.e. a 1:1 ratio, with co-culture for 24 hours, as well as, separately, for Treg/CD4+ T cells*
*(3:1 ratio, co-culture for 5 days). The authors need to clarify how and why these*
*particular conditions were chosen - presumably through a process of optimisation for*
*the proposed assessments? Have the authors used the same conditions previously in*
*other (published or unpublished) studies? Please clarify.*

For cytokine analysis assays we used a E:T ratio of 1:1 and 24 hours based on
optimisation assays. In revision this information is included in methods lines 651-
653 and Supplementary Figure 13. For 5-day culture stimulations for Treg/CD4 T
cells, we used previously published culture conditions (Chan et al, Cell Reports
Med, 2020). This information is now included in methods lines 651-653.

*4. The in vitro part of the authors' study focused on two types of innate immune cells in*
*their assessments of cellular responses to pRBC, namely (classical, CD14+) monocytes*
*and gd T cells. A third type of innate immune cells present in PBMC, namely NK cells,*
*have long been thought to play a pivotal role in the human innate immune response to*
*P. falciparum, begging the question of why the authors chose to neglect them in their*
*study. Here again, I feel, they should justify their choice for greater clarity.*

In original submission, we focused on monocytes and $\gamma\delta$ T cells as these have
been reported to be the major cell source of inflammatory cytokines that
contribute to severe malaria (Stanisic et al, I&I, 2014, ref #17). This is now
clarified in lines 74-76. In addition, we have previously shown that NK cells do
not robustly produce cytokines to un-opsonized parasites in similar in vitro
systems (Ty et al, Sci Trans Med, 2023, ref#25). Nevertheless, we agree that NK
cells are also important mediators of the innate response to malaria. As such, in
revision, NK cell responses have been included in Supplementary Figure 8B-C.
We did not detect any age dependent NK cell responses to parasites, and
cytokine production in response to parasite stimulation as negligible, consistent
with our previous study. In revisions, NK cell responses are introduced in lines
7274 and 85-87, results indicated in lines 298-302.

*Results & Discussion*

*1. With respect to the data generated on monocytes subjected to pRBC stimulation in*

*vitro:*

*(i) intracellular staining, as depicted in Figure 2B, revealed a significantly higher % of*
*adults' monocytes with CCL2, and a significantly higher % of childrens' monocytes with*
*IL10. Otherwise, with intracellular staining, there were no differences concerning the %*
*monocytes with any of the other pro-inflammatory cytokines assessed by this method*
*(IL1b, IL6 or TNF). At the transcriptional level, however, the picture appears quite*
*different. Here, as depicted in Figure 3G, both IL1b and TNF genes are highly*
*upregulated in adults' monocytes compared to childrens', along with several other*
*cytokine genes that showed a similar profile, but apparently not CCL2, results that*
*clearly do not align with those reported from intracellular staining. The transcriptional*
*profile of IL10 did reveal upregulation to a greater extent in childrens' monocytes, so*
*consistent here with the results obtained through intracellular staining. Somewhat*
*counter-intuitively, however, given its known association with IL10 production, Figure 3E*
*depicts the macrophage alternative activation signaling pathway as being upregulated in*
*adults' monocytes but - in counterpoint to the observations mentioned above -*
*downregulated in childrens'. The latter observation appears itself to contrast directly with*
*the finding (Lines 242-243), in childrens' monocytes, of the upregulation of*
*transcriptional factors, that included HOXA3, which inhibit M1 polarization and drive M2*
*(alternatively activated) polarization in monocytes. The interpretation of these*
*apparently contrasting data, including the relative weight given to the results of*
*intracellular staining versus those from transcriptional profiling, seems to me not so*
*clear cut as the authors' statement (Lines 255-257) indicates: 'transcriptional and*
*cytokine production data show that classical monocytes' responsiveness to malaria is*
*age dependent, with enhanced inflammatory responses in adults compared to children*
*in malaria naive individuals.' The disparities in the findings from the different data sets,*
*as are pointed out here, suggest to me that a rather more nuanced interpretation is*
*warranted than the one the authors have presented.*

In interpreting our data, we have attempted to holistically consider all
complementary data in our conclusions. We agree that numerous pathways
identified transcriptionally are of potential interest, and have as such tempered
language throughout this section, and included the emphasis that specific
validation is required to link transcriptional and phenotypic changes to functional
responses (see lines 283-285, 330-333, 460-462).

*(ii) There are other aspects of the data, in particular those concerning childrens'*
*monocyte activity, that merit some attention, currently missing, in the discussion, in my*
*opinion. It is striking, for example, that the basal (uRBC condition) % of childrens'*
*cytokine containing cells, with the exception of CCL2, was markedly higher than that of*
*adults (Figure 2A). In addition, as the authors point out, it was notable that childrens'*
*monocytes had a greater propensity for polyfunctionality in terms of their content of*
*more than 1 cytokine following pRBC stimulation (Figure 2C,D). Could the ex vivo*
*analyses the authors performed shed any light on this particular aspect e.g. differences*
*in the ex vivo activation status of childrens' versus adults' cells? Childrens' classical*
*monocytes did, for example, display significantly higher levels of CD86 than did adults'*
*monocytes (Supplementary Figure 2G), with the same being true for their different DC*

*populations. Notable also was the significantly higher level of expression of ICOS on*
*childrens' CD4+ T cells (Supplementary Figure 2I). These indications of apparently*
*higher levels of (pre-existing) activation of different cell types in children warrant some*
*discussion.*

We agree with Prof Luty that baseline differences could influence cytokine
production. In response to this query (as also raised by R2), we have included
new analysis of our data to explore correlations between baseline frequencies
and activation signatures of cells with cytokine responses in monocytes and Vδ2+
γδ T cells. Data is included in Supplementary Figure 9. This analysis identified
age- and cell type-specific associations between baseline immune cell
frequencies or activation states and cytokine responses to *Pf*, however most
associations were not significant. Notable associations included IL10 production
by CD14 monocytes with baseline activation of cDCs and pDCs in children and
TNF production by Vδ2+ γδ T cells with baseline CD16 responses in B cells, CD4
T cells, cDCs, pDCs and monocytes. These data are now reported in results
lines 319-330, and discussion lines 478-483. In addition, the importance of
identifying cell intrinsic and extrinsic factors under pinning our findings is noted in
lines 536-540.

*(iii) Following on from the above, from a purely practical/technical perspective, the*
*culture conditions employed would, I conjecture, have allowed for schizont rupture at a*
*certain point, and hence exposure of PBMC to the full array of parasite-derived*
*products, obviously including hemozoin, DNA, extracellular vesicles, as well as to intact*
*free merozoites, all of which could be expected to influence the outcomes measured via*
*the involvement of TLR-mediated interactions, as well as via other receptor/signaling*
*pathways. The authors should offer some comment on and discussion of these aspects,*
*whilst also considering the possibility that trained immunity might be playing a role here.*
*Such a phenomenon could potentially explain the apparently age-related upregulation of*
*the expression of pRBC-mediated TLR4 (higher in adults' monocytes) identified in this*
*study i.e. the possibility that adults' responses to pRBC in vitro might be skewed due to*
*prior/life-long exposure to other pathogens.*

Regarding schizont rupture in culture conditions, in all stimulation's, parasites
were from glycerolyte preserved trophozoite stages (methods line 636-637). As
such, these parasites are now non-viable and are unable to develop and rupture.
We apologise that these experimental conditions were unclear in first submission
and now included these clarifications in methods line 646-647.

We agree that trained immunity is an important potential mechanism that may
underpin our findings and have now included this concept in discussion
paragraph lines 460-477

*(iv) Figure 3 is entitled 'Increased transcriptional activation in response to P. falciparum*

*in monocytes from malaria naive adults compared to children', whilst Supplementary*
*Figure 5 is entitled 'IPA analysis of monocyte genes which were higher in children*
*compared to adults after primary pRBC stimulation'. For the uninitiated reader, and to*
*avoid confusion in interpretation, I suggest that the authors modify the latter title to read,*
*for example, 'Pathway and upstream regulator analyses of monocyte genes which were*
*higher in children compared to adults after primary pRBC stimulation'.*

We have made this title change as suggested.

*(iv) Figure 3H appears to lack the data relating to expression of one of the genes (Fas?)*
*in childrens' cells*

Figure 3H has expression of 4 genes, CD28, TLR4, TYROBP, FAS, each with
children (red) and adult (blue) data. We believe the confusion may stem from the
small size of the red bar representing children responses for CD28 at the top of
the Figure 3H leading to misinterpretation of the FAS red bar (children response)
as corresponding to TYROBP. In the revised version, we have made slight edits
to these figures so there is now a light grey line running between the gene sets,
rather than between the children and adult responses as in original submission.
We hope that will improve clarity of the figures in this section.

*2. In Lines 364-373 the authors discuss the potential role for different TLR in the pRBC-*
*induced monocyte responses measured. Whilst they correctly mention TLR3, TLR4,*
*TLR7 & TLR8, they fail to mention TLR9 in the same context. There is evidence in the*
*published literature identifying specific interactions between TLR9 and complexes of*
*hemozoin/parasite-derived DNA. This omission should be corrected for completeness.*

We agree with the reviewer that TLR9 is of potential importance and have now
included in discussion lines 416-419.

*3. In comparison to the data pertaining to monocyte activity, as discussed above, those*
*pertaining to gdT cell activity are less contentious in the sense that childrens' and adults'*
*cells responded to pRBC stimulation in broadly similar ways, albeit with significantly*
*higher % of adult cells containing IFNg and/or TNF. There is a wealth of evidence in the*
*literature supporting a major role for gd T cells in the innate immune response to P.*
*falciparum that these data serve to underscore. The data concerning CD4+ T cells (Treg*
*and others) were unremarkable in the sense that, under the chosen in vitro conditions,*
*childrens' and adults' profiles were very similar. In the context of the latter, the authors*
*might comment on the biological relevance of a pRBC:PBMC ratio of 3:1.*

While the biological relevance of in vitro stimulations is hard to assess, the ratios
of pRBC:PBMCs used in our assays are equivalent to 4 or 4.4 log₁₀ parasites/ul.
(Calculation is based on healthy male range of RBC concentration of 6mil/ul, and

a WBC concentration of 10000/ul resulting in a ratio of 600RBC:1WBC. A 1:1
ratio is thus equivalent of 0.16% parasitemia, or 10000parasities/ul, and a 3:1
ratio is equivalent to 0.5% parasitemia or 30000parasites/u). This information is
now included in methods lines 653-655 and 663-665. These concentrations of
parasites are within the range of that seen in our study participants during
infection (Figure 1A).

*4. Statements such as 'monocytes and Vδ2+ γδ T cells produced significantly higher
inflammatory cytokines in adults compared to children following in vitro parasite
stimulation' (Discussion lines 349-351, and several similar statements in Results) are
misleading in the sense that they imply - at least to me - the quantification of cytokines
in culture supernatants, whereas they are actually referring to the % cells detected as
positive by intracellular staining and/or transcriptional profiles. For clarity, these
statements should be reworded to better reflect the true nature of the data collected.*

As suggested to improve clarity we have made edits through results and
discussion as requested, see lines 176,178-179, 182, 238, 242, 291-298, 306,
669 317-318, 324-325, 398-400, 421, 429-431, 446.

*Conclusions*

*1. Whilst the in vitro studies performed here are clearly informative, although
nevertheless tempered by the specific co-culture conditions chosen, I think it is
important for the authors to point out - possibly in their conclusion and/or in the
paragraph on their study limitations - that primo-exposures in vivo, whether in children
or in adults, will necessarily involve initiation of immunological responses by pivotal
early-responding cell types, such as dendritic cells, that were not the focus of the study
here. Evidence supportive of a role for DC in mediating responses to asexual blood
stages can be found, for example, in the published literature emanating from in vivo
(CHMI) studies of primo-infections with P. falciparum in naive adults (e.g. Teirlinck et al,
Infection & Immunity, 2015).*

We agree that other cell types, specifically DCs, and cell extrinsic factors play
pivotal roles in the immune response and are not investigated in our study.
These limitations are now discussed in lines 570-576.

We thank the reviewers and editors for their time and insights. We have addressed the additional comments as outlined below. Page numbers indicated in the rebuttal are for track changes version of resubmission.

REVIEWERS' COMMENTS

Reviewer #1 (Remarks to the Author):

I thank the authors for their considerate responses to my queries, which I feel are now sufficiently addressed.

I have one very small new point, which the authors could consider revising (or not) in a final version, but would not require further review: the IPTi trial by Muhindo et al. quoted by the authors in the Discussion compared two different frequencies of DHA-PPQ IPTi (every 4 vs every 12 weeks), rather than IPTi vs no IPTi. The incidence of malaria episodes in the follow-up year after cessation of IPTi was lower in children who had received 4-weekly DHA-PPQ compared to those who had received it only 12-weekly. This still suggests that reducing the incidence of infection in younger children (e.g. by vaccination) has beneficial rather detrimental consequences in older children, but strictly speaking it cannot be surmised (as the authors seem to suggest) that children who had not received any IPTi would have had a(n even) higher incidence of malaria during this follow-up year than the (12-weekly) IPTi recipients.

We have updated the sentence to increase clarity (Lines 496-498).

However, clinical trials of infants who received dihydroartemisinin-peperaqueine to prevent malaria between 8 weeks to 24 months of age, showed that children who received 4 weekly doses (but not 12 weekly doses) of chemoprevention had reduced episodes of malaria in the year following drug cessation⁸³.

Reviewer #2 (Remarks to the Author):

I think the authors have sufficiently addressed all my comments and I have no further queries regarding the manuscript.

Reviewer #2 (Remarks on code availability):

The code is well annotated and easy to read with no obvious errors. The readme contains information about packages and versions. However, I did not install and run the code to look for errors.

We thank the reviewer for their time and comments.

Reviewer #3 (Remarks to the Author):

The revised version of the manuscript has taken into account my comments largely to my satisfaction.

I have just one outstanding issue concerning methodology that the authors should address:

As they will surely know, cryopreservation methods exist that would have allowed for the authors to have used pRBC containing live, viable parasites for in vitro stimulations as opposed to the non-viable pRBC resulting from the glycerolyte-based method chosen. The use of viable parasites would, I suggest, have more authentically replicated in vivo conditions. Do they know, or can they at least conjecture, what the condition was (ruptured/lysed/intact throughout?) of the non-viable pRBC they used over the period of culture in either 1- or 5-day stimulations? The reasons for the choice they made in this context should be explained for clarity, and the fact that they used non-viable pRBC for the stimulations should also be included as a further limitation of their study.

Cryopreserved parasites were used to allow for a single batch of parasites to be used across all assays in this study. This is now indicated in lines 669 of methods. Thawed parasites were intact late stage trophozoites. The use of cryopreserved parasites is in line with previously published studies cited in methods.

While we did not formally assess the state of the parasites after a 1 or 5 day culture, these parasites would be unlikely to have ruptured as they are non-viable post cryopreservation. Intact parasites would however been uptaken by phagocytes in culture conditions. We have added a limitation of non-viable parasites in the limitations as requested (lines 561-564).